# Quantifying projected changes in runoff variability and flow regimes of the Fraser River Basin, British Columbia

Siraj Ul Islam[1*], Charles L. Curry[2], Stephen J. Déry[1], and Francis W. Zwiers[2]

[1]Environmental Science and Engineering Program,
University of Northern British Columbia, Prince George, British Columbia (BC), V2N 4Z9, Canada

[2]Pacific Climate Impacts Consortium,
University of Victoria, Victoria, BC, V8W 2Y2, Canada

*Correspondence to*: Siraj Ul Islam (sirajul.islam@unbc.ca)

**Abstract.** In response to ongoing and future-projected global warming, mid-latitude, nival river basins are expected to transition from a snowmelt-dominated flow regime to a nival-pluvial one with an earlier spring freshet of reduced magnitude. There is, however, a rich variation in responses that depends on factors such as the topographic complexity of the basin and the strength of maritime influences. We

illustrate the potential effects of a strong maritime influence by studying future changes in cold season flow variability in the Fraser River Basin (FRB) of British Columbia, a large extratropical watershed extending from the Rocky Mountains to the Pacific Coast. We use a process-based hydrological model driven by an ensemble of 21 statistically downscaled simulations from the Coupled Model Intercomparison Project Phase 5 (CMIP5) following the Representative Concentration Pathway (RCP)

8.5.

Warming under RCP8.5 leads to reduced winter snowfall, shortening the average snow accumulation season by about one-third. Despite this, large increases in cold season rainfall lead to unprecedented cold season peak flows and increased overall runoff variability in the VIC simulations. Increased cold season rainfall is shown to be the dominant climatic driver in the Coast Mountains,

contributing 60% to mean cold season runoff changes in the 2080s. Cold season runoff at the outlet of the basin increases by 70% by the 2080s and its interannual variability more than doubles compared to the 1990s, suggesting substantial challenges for operational flow forecasting in the region. Furthermore, almost half of the basin (45%) transitions from a snow-dominated runoff regime in the 1990s to a

5   primarily rain-dominated regime in the 2080s according to a snowmelt pulse detection algorithm. While these projections are consistent with the anticipated transition from a nival to a nival-pluvial hydrologic regime, the marked increase in FRB cold season runoff is likely linked to more frequent landfalling atmospheric rivers in the region projected in the CMIP5 models, providing insights for other maritime-influenced extratropical basins.

**Keywords:** Climate change, hydrological modelling, runoff, snow, mid-latitude basin, Fraser River Basin

# 1 Introduction

Rising air temperatures and changes in precipitation patterns are altering hydrological processes and states in river basins across the globe, including those in cold regions. Snow-dominated (nival) river basins are particularly sensitive to warming air temperatures as these can lead to marked decreases in seasonal and longer-term water storage that otherwise provides a reliable source of streamflow generation during the spring and summer melt periods (Barnett et al., 2005). This loss of storage is expected to lead to increased interannual streamflow variability (Fleming and Clarke 2005, Fountain and Tangborn 1985). Increasing interannual variability has been observed in the flows of nival and glacial rivers across Canada in recent decades (Déry et al., 2009, 2012). It is not yet known, however, whether future changes in climate will further impact the interannual variability in flows for snow-fed rivers, as previous studies focused solely on changes to mean annual or seasonal flows (e.g. Shrestha et al., 2012; Gelfan et al., 2017, MacDonald et al., 2018).

Warming is also expected to affect streamflow seasonality and the hydrological regimes of nival basins. The timing of the spring freshet advances with warming and a decline in the seasonal snowpack, leading to earlier summer recession (Hodgkins and Dudley, 2006; Moore et al., 2007; Kang et al., 2016). As a result, hydrographs exhibit earlier rising limbs that signal spring snowmelt flow enhancement and earlier annual centres of flow volumes. Such hydrological regime shifts have been observed in nival basins across North America (Stewart et al., 2005; Burn, 2008; Fritze et al., 2011) and Eurasia (Tan et al., 2011). In a projected warmer climate, such regime shifts are likely to intensify (e.g. Stewart et al., 2004; Schneider et al., 2013; Islam et al., 2017) although regional responses may be quite

distinct owing to each basin's unique characteristics such as elevation range, permafrost distribution, continentality and proximity to the 0°C threshold air temperature during the cold season. The fine-scale geographical distribution of such flow regime shifts, however, is not well covered in the current literature, and could have implications for reginal adaptation measures and water resources

management.

In mid-latitude coastal basins with maritime influences, the projected hydrologic response to warming will be influenced by two possibly confounding factors, change of the snow-to-rain ratio and a substantial increase in the atmospheric moisture supply (largely via atmospheric rivers, ARs) to the coastlines of western North America (Payne and Magnusdottir, 2015; Radic et al., 2015; Warner et al.,

2015; Warner and Mass, 2017; Curry et al., 2019) and Europe (Lavers and Villarini, 2013; Lavers et al., 2015). Historically, ARs, mainly in the cold season, have been linked to extreme precipitation and flooding in basins located on the periphery of the west coast of North America (Ralph et al., 2006; Neiman et al., 2008; Guan et al., 2010; Dettinger et al., 2011). Consequently, future increases in the frequency and intensity of ARs (e.g., Espinoza et al., 2018) may induce much larger seasonal flows in

the basins exposed to AR paths, particularly those on the eastern boundaries of the Pacific and Atlantic Oceans.

British Columbia's Fraser River Basin (FRB) forms one of the largest nival watersheds draining the western Cordillera of North America (Benke and Cushing, 2005) (Fig. 1). It is a mid-latitude, mountainous basin in western Canada in proximity to the eastern Pacific Ocean where a prominent

centre of AR activity exists (Guan and Waliser, 2015; Radic et al., 2015; Gershunov el., 2017). Recent studies focused on the basin have reported observed and projected changes in runoff timing and

magnitude in the FRB under changing climate (Shrestha et al., 2012; Kang et al., 2014, 2016; Islam et al., 2017). Using the mean climatological hydrograph, these authors noted an advance of the spring freshet and reduced summer peak flow in the FRB's major tributaries and the main stem of the Fraser River. In contrast, little attention has been focused on the detection of changes in interannual and daily flow variability and cold season flow magnitude in the Fraser River and its tributaries. Assessing precisely how the FRB's flow variability will change in a warmer climate requires the use of advanced downscaling methods and simulation tools along with improved future climate projections. In particular, these modelling tools can help us understand how projected changes in flow variability are related to changes in the proportionality of snowfall to rainfall and snowmelt-driven runoff timing. Changes in flow variability may increase the potential for flooding, thus threatening the natural, ecological and social systems within the river basins.

The principal goals of this study are therefore: 1) to investigate how projected climatic change affects the mean state and daily time scale variability of FRB flows, 2) to illustrate the potential effects of a strong maritime influence on cold season flows that is punctuated by more frequent, intensifying ARs, and 3) to evaluate the likelihood of transitions from snowmelt-dominant to hybrid snowmelt-rainfall or rainfall-dominant flow regimes in a spatially explicit manner across the basin. The latter is achieved via the use of a snowmelt pulse detection technique to distinguish distinct runoff regimes, applied within a semi-distributed, macroscale hydrological model driven by 21 downscaled simulations of future climate from Global Climate Models (GCMs). This approach provides insight into the location and timing of these transitions, while the use of many different GCM-driven hydrological simulations allows a concomitant estimate of the associated uncertainties. The present work is the first of two papers

analysing the same set of hydroclimatic simulations. The present effort deals with features of the transition from seasonal snow to a hybrid snowmelt/rainfall runoff regime, with special attention to the changes in snowmelt dynamics and daily runoff variability. A companion paper (Curry et al., 2019) addresses the consequences of these changes for river discharge, including a formal flood frequency (extreme value) analysis for the 21$^{st}$ century. In nival basins that are exposed to AR activity, the impact of projected changes in ARs frequency and intensity on streamflow variability and hydrologic regimes has not yet been extensively studied. These two papers address this question specifically for the FRB, but also demonstrate a framework for assessing such impacts in other basins.

## 2. Domain, Modelling Framework, and Methodology

### 2.1 Study Domain

Our primary focus is on the FRB (Fig. 1a) that extends from the Pacific coast to the continental interior, and spans 240,000 km$^2$ of diverse landscapes including dry interior plateaus bounded by the Rocky Mountains to the east and the maritime influenced Coast Mountains in the west. Its elevation ranges from sea level to 3954 m at its tallest peak, Mt. Robson in the Rocky Mountains (Benke and Cushing, 2005). Descending at Fraser Pass near Blackrock Mountain, the Fraser River runs 1400 km before draining into the Strait of Georgia and Salish Sea at Vancouver, British Columbia (BC). Over the past 60 years, mean annual surface air temperature in the FRB has risen by 1.4°C modifying the FRB's natural water cycle (Kang et al., 2014). Impacts of this warming include reductions in snow accumulation (Danard and Murty, 1994), declines in the contribution of snow to runoff generation (Kang et al., 2014) and earlier melt-driven runoff with subsequent reductions in summer flows (Kang et

al., 2016). The corresponding changes in mean flow (Danard and Murty, 1994; Morrison et al., 2002; Ferrari et al., 2007; Kang et al., 2014, 2016) have been accompanied by considerable amplification in the interannual variability over recent decades across many streams and rivers in the FRB (Déry et al., 2012).

Our analysis focuses on the Fraser River main stem at Hope, BC (Lower Fraser, LF) since it integrates flows from about 94% of the FRB area, and is the location of the longest instrumental streamflow record for the basin's main stem. We also consider four mountainous sub-basins (the Upper Fraser (UF), Quesnel (QU), Chilko (CH) and Thompson-Nicola (TN) Rivers; Fig. 1a, Table 1), along with three geoclimatic regions (the Interior Plateau, the Rocky Mountains, and the Coast Mountains; Moore,
1991, Fig. 1b). These regions represent the range of distinctive physiographic and hydroclimatic conditions found within the FRB. The FRB exhibits snowmelt-dominant flows in all sub-regions in late spring and early summer in the current climate. In addition, several catchments in the Coast Mountains and in the LF exhibit a secondary runoff peak owing to Pacific synoptic storms often associated with ARs in October-December.

**2.2 Climate Models, Observational Data, and Statistical Downscaling**

We used statistically downscaled climate simulations from 21 GCMs (Supplementary Table 1) submitted to the Coupled Models Intercomparison Project Phase 5 (CMIP5) (Taylor et al., 2012). A single realization was used from each GCM, driven by historical greenhouse gas and aerosol forcing up to 2005 and Representative Concentration Pathway 8.5 (RCP 8.5) forcing subsequently. GCM-
simulated daily precipitation and daily maximum and minimum surface air temperature for the period 1950-2099, downscaled to 10-km spatial resolution, were obtained from the Pacific Climate Impacts

Consortium (PCIC). Downscaling is necessary to apply the coarse-scale GCM results (ranging from 1.1° to 3.7° in longitude and 0.9° to 2.8° in latitude; Supplementary Table 1) at the finer scale of the hydrologic model, which is configured at 0.25° horizontal resolution in both latitude and longitude. The downscaling process also corrects GCM biases in air temperature and precipitation relative to the

ANUSPLIN station-based daily gridded climate dataset (Hopkinson et al., 2011; NRCan, 2014). ANUSPLIN refers to the gridding technique, which is based on the Australian National University spline interpolation method (Hutchinson et al., 2009). This dataset contains gridded daily maximum and minimum air temperature (°C) and total daily precipitation (mm) data for the Canadian landmass at a spatial resolution of 0.0833° (~10 km × 10 km, depending on latitude).

PCIC performed the downscaling with the Bias Correction/Constructed Analogue Quantile Mapping method, version 2.0 (BCCAQ2), a hybrid approach that combines the bias corrected constructed analogues (BCCA; Maurer et al., 2010) and bias corrected climate imprint (BCCI; Hunter and Meentemeyer, 2005) techniques. BCCA bias-corrects the large-scale daily GCM temperature and precipitation using quantile mapping to a target gridded observational product (here, ANUSPLIN),

aggregated to the GCM grid-scale and then integrates spatial information by regressing each daily large-scale temperature or precipitation field on a collection of fine-scale historical analogues selected from ANUSPLIN. Using this relationship, fine-scale patterns are generated from the target dataset. In parallel, BCCI applies quantile mapping to daily GCM outputs that have been interpolated to the high-resolution grid based on "climate imprints" derived from long-term ANUSPLIN climatologies (Hunter

and Meentemeyer, 2005). BCCA produces results that may be subject to insufficient temporal variability whereas BCCI can contain artifacts due to spatial smoothing. In BCCAQ, daily values within

a given month from BCCI are reordered according to their corresponding ranks in BCCA, improving the spatiotemporal variability (Werner and Cannon, 2016). BCCAQ2 further refines BCCAQ by substitution of quantile delta mapping instead of regular quantile mapping in BCCI to preserve the magnitude of projected changes over all quantiles from the GCM in the downscaled output (Cannon et al., 2015; Li et al., 2018). The performance of the bias correction method depends mainly on the target dataset used for corrections. In this respect, the known biases of ANUSPLIN (e.g., the low precipitation bias ~2-5 mm day$^{-1}$ at high elevations compared to other datasets; Islam et al., 2017) are transmitted to the downscaled model results via BCCAQ2.

By better representing historical daily variability and also the antecedent drivers of daily streamflow extremes, BCCAQ2 represents an advance over the Bias Correction and Spatial Downscaling (BCSD) method used in Islam et al. (2017). While BCSD reflects historical intra-month variability via stochastic sampling, BCCAQ2 preserves climate model skill in simulating daily variability, where and when it exists.

## 2.3 Variable Infiltration Capacity (VIC) Model and Simulation Strategy

We used the semi-distributed macroscale Variable Infiltration Capacity (VIC) model (Liang et al., 1994, 1996) to simulate hydrological processes in the FRB. The VIC model has been extensively used for climate change research over various river basins (e.g. Adam et al., 2009; Cuo et al., 2009; Hidalgo et al., 2009; Elsner et al., 2010; Gao et al., 2010; Wen et al., 2011; Schnorbus et al., 2011; Zhou et al., 2016) including the FRB (Shrestha et al., 2012; Kang et al., 2014, 2016; Islam et al., 2017; Islam and Déry, 2017). It conserves surface water and energy balances for large-scale watersheds such as the FRB (Cherkauer et al., 2003). VIC simulates the sub-grid variability by dividing each 0.25° × 0.25° grid cell

into several elevation bands (Nijssen et al., 2001), each of which is further subdivided into a number of tiles that represent different land surface types, producing a matrix delineated by topography and land surface type. Energy and water balances and snow are determined for each tile separately (Gao et al., 2009). The VIC model is coupled to a routing scheme (Lohmann et al., 1996, 1998a, b) that approximates the runoff from gridcells using a known channel network (Wu et al., 2011). Streamflow produced in this way is extracted at outlet points of specific sub-basins of interest (Lohmann et al., 1996, 1998a, b).

After the CMIP5 projections were bias-corrected and downscaled to the ANUSPLIN grid, the resulting fields were averaged to a resolution of $0.25° \times 0.25°$ to match the VIC model grid. In addition to the daily meteorological forcings mentioned in Sec. 2.2, VIC also requires daily wind forcing and a number of static gridded fields (as reported in Kang et al., 2014) to characterize soil type, vegetation type, and elevation. The wind fields were obtained by interpolating coarse-scale ($2.5° \times 2.5°$) daily wind speeds at 10-m height above ground from the National Centers for Environmental Prediction/National Center for Atmospheric Research (NCEP/NCAR) Reanalysis (Kalnay et al., 1996) to the VIC grid.

Calibration and validation of the VIC model for this study was conducted through retrospective hydrologic simulations from 1979 to 2006 using ANUSPLIN re-gridded to the 0.25° horizontal resolution of VIC. The model was run in water balance mode using a daily time step, 3 soil layers and 10 elevation bands in each gridcell. Model parameters were calibrated based on an optimization process that minimizes the difference between observed and simulated hydrographs using the Nash–Sutcliffe efficiency (NSE) coefficient (Nash and Sutcliffe 1970). The daily NSE values for the entire FRB and its sub-basins range between 0.64 to 0.90 in the calibration time period revealing reliable application of the

VIC model (see Table 2 in Islam et al., 2017). Figure 2 describes the full experimental setup while further details of the VIC model implementation and application to the FRB are described in Islam et al. (2017) and Islam and Déry (2017).

The calibrated VIC model was integrated at a daily time step from 1950 to 2099 using statistically downscaled daily precipitation, maximum and minimum air temperature from each of the 21 downscaled CMIP5 GCM simulations (using historical and RCP 8.5 forcing) at 0.25° horizontal resolution. We initialized each simulation by running VIC for five years using the 1950 meteorological forcings to allow the model to spin-up. 30-year time periods were analysed in detail: the 1990s (1980 to 2009), 2050s (2040 to 2069) and 2080s (2070 to 2099) and changes relative to the 1990s were described.

## 2.4 Analysis Methods

### 2.4.1 Analysed Variables

Our analysis focuses primarily on VIC-simulated daily values of total runoff computed as the sum of baseflow and surface runoff at each model gridcell (VIC does not simulate sub-surface flows between gridcells). At the basin outlet, routed streamflow is converted to areal runoff by division of the corresponding basin area and then converting to precipitation equivalent units. To support our results of projected changes in runoff, we evaluate several other variables such as total precipitation, snowfall, rainfall, air temperature, and VIC-simulated snow water equivalent (SWE) and snowmelt. VIC uses specified air temperature thresholds to determine precipitation phase. In the current implementation, total precipitation is classified as 100% rainfall for temperature above 1.5°C, 100% snowfall for

temperatures below -0.5°C and is partitioned as a mixture of rainfall and snowfall for temperatures between these two thresholds.

We perform analyses using the water year, defined here as 1 October to 30 September of the following calendar year. The cold season is defined as the period from 1 October to 31 March, except for the cold season peak flow analysis, where the end of the cold season is taken to be the day when the 3-day running mean daily air temperature first exceeds 0°C at each gridcell between 1 and 31 March. Using the additional condition based on air temperature helps to identify the end of the cold season more precisely in each year. The last day of the cold season therefore depends on the air temperature criterion. A 3-day running mean avoids events when daily mean temperature suddenly exceeds 0°C for only a single day within the cold season. The snow accumulation season comprises days of the cold season when daily SWE accumulation, the difference between daily snow accumulation and snow melt, exceeds 1 mm.

## 2.4.2 Mean and Variability Calculations

Considering each model realization as a valid approximation to the real climate, we use the multi-model ensemble (MME) results summarised with four statistics: the temporal mean and standard deviation of each model, and the multi-model mean and inter-model standard deviation. Specifically, for a daily variable $x$ and model $i$, the mean climatology $\bar{x}_i$ and the interannual standard deviation $\bar{S}_i$ is calculated for each 30-year analysis period (i.e. 1990s, 2050s, and 2080s). The MME means of these quantities, $\bar{x}$ and $\bar{S}$, are then estimated, and the inter-model spread, $S_{MME}$ is characterized by the 5-95% models range in $\bar{x}_i$.

The effect of transient warming within the 2050s and 2080s periods in the RCP8.5 scenario is removed by subtracting the least squares linear trend from each time series before calculating its variability. Variability in 7-day runoff is computed using a 7-day running mean of the daily runoff time series.

In an attempt to better understand the contributions of future air temperature and precipitation change to runoff alteration in the simulations, we use a multivariate linear regression (MLR) analysis, following a similar approach by Kapnick and Delworth (2013). We decompose the cold season runoff $Rf$ into separate contributions from rainfall $Rn$, snowfall $Sn$, mean air temperature $T$ and residual $E$ on a monthly basis as follows:

$$\frac{\Delta Rf_{n,m}}{Rf_{m,1990s}} = a\frac{\Delta Rn}{Rf_{m,1990s}} + b\frac{\Delta Sn}{Rf_{m,1990s}} + c\frac{\Delta T}{Rf_{m,1990s}} + E \qquad (1)$$

where $a$, $b$ and $c$ are regression coefficients corresponding to the rainfall, snowfall and mean air temperature, $m = 1, \dots, 6$ denotes the cold season month (Oct-Mar), and $n = 2070, \dots, 2099$ represents the water year. $Rf_{m,1990s}$ represents the multi-year mean runoff for each month in the 1990s time period. The regression model is fitted for each gridcell and model independently using detrended monthly anomalies from 2070-2099. Spatial averages over the FRB and geoclimatic regions only use gridcells for which the MLR is statistically significant with a p-value $< 0.05$. The relative contribution of each variable to future runoff change is obtained by normalizing the area average of each term in Eq. (1) by the corresponding area-averaged $\Delta Rf$. We only consider the lag-0 correlation between the driving and response variables considering that monthly time resolution is sufficient to encompass any lags at the local grid scale.

## 2.5 Snowmelt Pulse Detection

Apart from the analysis of changes in cold season flow variability, we estimate the flow regime transitions that are usually induced by snowpack reduction under increasing summer temperatures. We investigate transitions to new hydrological regimes in the FRB using the Snowmelt Pulse (SP) detection technique (Cayan et al., 2001; Fritze et al., 2011). This technique separates snowmelt-dominant flows from rainfall-dominant flows using the maximum cumulative flow departure from mean flow within the defined time window. The SP date, which is defined as the day when the cumulative departure from that water year's mean flow is most negative, provides a way for determining the time at which increased ablation in a snowmelt-dominant basin initiates the transition from low winter base flows to high spring flow (freshet) conditions. While accumulation of flow departures for each water year commences on 1 October, we only consider those SPs occurring between 1 March (water-day 151; 152 for leap years) and 15 June (water-day 238; 239 for leap years), the present-day freshet period, so as to exclude runoff pulses induced by rainfall events. Runoff is rainfall-dominant when the ratio of the area of positive cumulative flow departure (indicating rainfall events) to the area of negative departure between water-days 151 and 238 is greater or equal to unity. An illustration of the robustness of the algorithm to different river flow regimes using historical data is shown in Supplementary Fig. 1. The application of the algorithm reveals the presence of a SP in the snowmelt-dominant system in all selected years, while for the rainfall-dominant system, SPs are quite rare.

To explore potential regime changes in the FRB, we evaluate and compare the fraction of years for which SPs are recorded within each analysis period (1990s, 2050s, 2080s) at each 0.25° gridcell and for all CMIP5-VIC simulations (with sample size of 21 CMIP-VIC simulations $\times$ 30 years = 630 years).

In each simulation, each gridcell is classified into one of four snow-dominant categories (SDCs) as defined by Fritze et al. (2011): SDC1 (clearly rain-dominant: SP occurrence in < 30% of water years); SDC2 (mostly rain-dominant, SP occurrence ≥ 30% but < 50%); SDC3 (mostly snowmelt-dominant, SP occurrence ≥ 50% but < 70%); and SDC4 (clearly snowmelt-dominant, SP occurrence ≥ 70%). This allowed spatial comparison of regime projections for the 2050s and 2080s with the regime characteristics of the 1990s. Finally, the SDC results are aggregated by geoclimatic region.

Overall, this study expands on Islam et al. (2017) who used 12 driving GCMs and only considered projections up to the 2050s to quantify changes in the FRB's mean runoff. Here we evaluate projected changes in runoff variability and flow regimes by the end of this century utilizing a set of 21 CMIP5 GCMs downscaled and bias-corrected using an advanced BBCAQ2 method and an efficient snowmelt pulse detection algorithm.

## 3 Results

We first examine the projected changes in the mean and interannual variability of precipitation over the different geographic regions of the FRB. Next, we explore the consequences of these changes for runoff means and variability at various temporal and spatial scales and estimate the contribution of key drivers that control changes in runoff mean. This is followed by a discussion of changing flow regimes over the FRB at regional and sub-basin scales.

### 3.1 Projected Changes in Precipitation and Snow-to-Rain Ratio

The MME mean precipitation, spatially averaged over the FRB, rises steadily over the simulation period, increasing nearly 15% in the 2080s relative to the 1990s in the cold season (Fig. 3a). The

changes are largest in the northern and eastern FRB (Supplementary Fig. 2a, c) reaching up to 20% (Supplementary Fig. 2c). The MME mean precipitation interannual variability increases by 15% with warming between 2 and 5 °C, then increases more sharply to over 25% as this level of warming is exceeded (after the 2060s) (Fig. 3a). The cold season precipitation variability increases approximately

linearly at a rate of ~4% °C$^{-1}$ towards the end of the 21$^{st}$ century compared to the 1990s, about double the rate of change of MME mean precipitation. The larger increase in precipitation variability compared with mean precipitation is seen throughout the simulation period for both the entire FRB and its three geoclimatic regions. Nevertheless, the models' 5-95% range tend to overlap (except in the Interior Plateau; Fig. 3e), reflecting the considerable spread amongst models.

The partitioning of MME mean total precipitation into rainfall and snowfall reveals substantial increases in daily rainfall towards the end of the 21$^{st}$ century across the Coast Mountains, Interior Plateau and Rocky Mountains (Fig. 4a, c, e). The increase in rainfall emerges prominently in the Coast Mountains in the latter half of the 21$^{st}$ century, especially in the cold season. Simultaneously, snowfall decreases (Fig. 4b) markedly in this region. Snowfall also decreases in the Interior Plateau and Rocky

Mountains (Fig. 4d, f), but to a lesser degree than the Coast Mountains, probably due to persistent cold temperatures at the higher elevations that dominate in this region (Table 1).

        Warming temperatures and reduced snowfall induce considerable changes in the snow accumulation and ablation seasons and in snowmelt (Fig. 5). Day-to-day SWE accumulation declines while its seasonality shifts over the 21$^{st}$ century, again with more prominent changes in the Coast

Mountains relative to other regions (Fig. 5a). The length of the snow accumulation season is about 38% shorter on average in the 2080s for all geoclimatic regions relative to the 1990s with a reduction from

nearly 80 to 50 days in the Coast (Fig. 5a) and Rocky Mountains (Fig. 5e), and from 65 to 40 days in the Interior Plateau (Fig. 5c). The magnitude and seasonality of snowmelt (Fig. 5b, d, f), which is responsible for generating high flows typically in May or June, shows earlier snowmelt freshets in the future and reduced snowmelt volume. Changes in the partitioning of precipitation between rainfall and snowfall greatly impact the seasonal SWE distribution, consistent with the findings of Islam et al. (2017). While snowmelt during the freshet diminishes overall in the future, unprecedented snowmelt events begin to appear during the cold season in the Coast Mountains (Fig. 5b) by the 2050s, likely due to more frequent warming episodes or perhaps increase in rain-on-snow events.

## 3.2 Projected Changes in Runoff Mean, Variability and Seasonality

Changes in cold season runoff (Fig. 3b) in the FRB are driven by both warming and increases in the mean and variability of precipitation (Fig. 3a). Consequently, the CMIP5-VIC simulations display larger increases in runoff variability than in mean runoff throughout the simulation period for the entire FRB, Interior Plateau and Rocky Mountains regions. This is not, however, the case in the Coast Mountains, where the increase in mean runoff (55% by the 2080s) is substantially larger than that in runoff variability (40%). This finding helps to explain why the increase in future runoff is much more evident in the Coast Mountains region (Fig. 6).

The substantial increases in cold season runoff are summarized by sub-region and sub-basin in Table 1. Of these, Thompson-Nicola exhibits the largest relative change (+140%) from the 1990s to 2080s, although it is historically the driest of the sub-basins, while the runoff at Hope increases by 71% between the same epochs. With respect to annual runoff, only the Coast Mountains and the Chilko sub-basin display substantial increases (but much smaller in relative terms than cold season increases), with

little change elsewhere (Supplementary Table 2). The same qualitative results hold for runoff variability as for means, both in the cold season and annually (Supplementary Table 2). In addition, by the 2080s, the runoff mean and standard deviation more than double over 83% and 71% of the FRB, respectively (Supplementary Fig. 3).

The future evolution of runoff seasonality shows that while the dominant snowmelt-generated peak flow shifts earlier by ~1 month, noticeable cold season runoff events emerge in winter and spring at the end of the 21$^{st}$ century (Fig. 6). This is most pronounced in the Coast Mountains where fall-winter runoff events rival the summer peak runoff in magnitude (Fig. 6a). The spatially-averaged runoff over the Coast Mountains further highlights the strong increase in cold season peak runoff in this region (Fig.
7). This increase in cold season peak runoff magnitude is simulated across the CMIP5-VIC ensemble (Fig. 7c). Apart from the increase in cold season peak runoff magnitude and its annual variability (Fig. 7a), the corresponding peak flow occurs somewhat later with warming in the Coast Mountains, moving from late November (~water day 50) to the beginning of December (~water day 60) at the end of the 21$^{st}$ century (Fig. 7b). Compared to the Coast Mountains, the changes in cold season peak flow timings
are much larger in the Interior Plateau and Rocky Mountains probably due to more frequent winter rainfall events on snowpacks (Fig. 4 c and e).

The MME projected hydrograph (routed streamflow) for the Fraser River at Hope (Fig. 8a) shows more runoff (estimated using the VIC routing scheme) in the late winter and spring owing to the earlier onset of spring snowmelt, which advances by nearly 25 days in the 2050s (consistent with Islam
et al. (2017)) and 40 days in the 2080s relative to the 1990s. The magnitudes of the annual peak and post-peak flows are, however, progressively diminished in the future periods, with reduced discharge

until early October. These changes indicate earlier recession to progressively lower flows in summer when salmon are migrating up the Fraser River.

Daily mean hydrographs (routed streamflow) in the Upper Fraser, Quesnel, Thompson-Nicola and Chilko sub-basins exhibit similar features of future change as those seen at Hope (Supplementary Fig. 4). The advance in the timing of the annual peak flow in these sub-basins is slightly less than for the FRB as a whole (Table 1), presumably due to their higher mean elevations and lower cold season temperatures. The Chilko features a later freshet by ~35 days in the base period compared to the other three sub-basins. This reflects the fact that its flow is partially controlled by the Coast Mountains with influence from the Pacific Ocean along with the presence of extensive glaciers in the basin. Possibly due to these factors, the Chilko sub-basin exhibits less of an advance in peak runoff in the future and only slightly reduced peak flow magnitude, unlike the other sub-basins and the FRB as a whole (Supplementary Fig. 4j and Fig. 8a).

The changes in daily runoff variability (interannual variability of each day of the water year) are modest in summer with small decreases that are consistent with corresponding runoff decreases (Fig. 8b). In contrast, the variability increases substantially in the cold season with greater increases in the 2080s than in the 2050s for the Fraser River at Hope (Fig. 8b). Similar changes also emerge in the Upper Fraser, Quesnel, Thompson-Nicola and Chilko sub-basins that exhibit increasing cold season variability with magnitudes comparable to the Fraser River at Hope (Supplementary Fig. 4). The changes in daily variability in 7-day moving windows of daily runoff are fairly large in the cold season (Fig. 8c) revealing increased day-to-day flow fluctuations along with an increase in the interannual variability of daily variability in the 2050s and 2080s.

### 3.3 Key Climatic Controls of Runoff in the CMIP5-VIC Simulations

The MLR analysis (as described in section 2.4.2) determines the contribution of key climatic drivers such as rainfall, snowfall and mean air temperature to the VIC simulated change in cold season mean runoff at each gridcell in the FRB. The response variables (rainfall, snowfall and temperature) used in the MLR equation, however, are almost certainly not independent given the direct effect of temperature on precipitation phase and snowmelt rate.

The regression model tends to overestimate the mean runoff change (Table 2) when compared with VIC simulations. Overall, however, the models appear to perform well by estimating general patterns of cold season runoff changes somewhat similar to that of the VIC simulated runoff changes over a large portion of the basin with explained variance ranges between 50-90% (Supplementary Fig. 5). While changes in snowfall contribute only 5 to 7% to the runoff changes, changes in both mean air temperature and rainfall are about equally influential for runoff change in the Interior Plateau and Rocky Mountains. In the Coast Mountains, the projected rainfall leads temperature and contributes 61% to the runoff change compared to a 32% contribution of temperature change (Table 2). This analysis further confirms the increasing cold season moisture supply (Fig. 4) and associated runoff increase (Fig. 6) in the Coast Mountains.

### 3.4 Changes in Runoff Regimes

Projected FRB flow regimes are assessed using the SP detection algorithm described in Sec. 2.5. SPs occur in the VIC simulations in all years at nearly all gridcells in the 1990s (Supplementary Fig. 6), resulting in an SDC4 flow regime classification throughout the basin except in the main stem Fraser

River downstream from Hope (Fig. 9). The frequency of SPs decreases in future periods with the maximum change in the Interior Plateau, where the SP years decrease to 43±7% in the 2080s when compared to 100% SP years in the 1990s. Such decreases in SPs are more modest in the Rocky (64±6%) and Coast Mountains (57±4%) (Table 3). A majority of gridcells in the Interior Plateau transitions from snowmelt-dominant SDC4 to rainfall-dominant SDC1 (33% of gridcells) or SDC2 (26%) by the end of this century. By contrast, the Rocky and Coast Mountains show resilience to regime transitions compared to lower elevation regions and remain mostly snowmelt-dominant SDC3 or SDC4 in the 2080s – see Table 3 for details. In the Coast Mountains, the higher elevations resist regime transitioning compared to other elevations that have robust transitions to rainfall-dominant SDC1 or SDC2 regimes. In contrast with the spatially-averaged response of the geoclimatic regions, routed flows at the outlets of the Upper Fraser, Quesnel, Thompson-Nicola, Chilko and Fraser River at Hope show a weaker transition of flow regimes (Table 3 and Supplementary Fig. 7). This characteristic arises from the VIC routing procedure, wherein model gridcells contributing flows to the outlet occupy mostly higher elevations and hence produce flow statistics with more SPs. This attenuation of the climate change signal at channel outlets is consistent with the recent findings of Chezik et al. (2017).

## 4 Discussion

Our results suggest that under the projected warming, the changes in precipitation variability and phase, as simulated by CMIP5 models (Pendergrass et al., 2017), play a leading role in declining cold season snowpack accumulation and shifting spring snowmelt earlier in FRB. Thus projected increases in the precipitation rain-to-snow fraction will have a strong impact on the severity of flooding, for example on mountainsides, with increased spring rainfall accelerating snowmelt runoff. Annual peak flows in the

Coast Mountains having a strong maritime influence will shift earlier by around one month by the 2080s and more frequent cold season runoff events will rival spring freshet flows in magnitude by the end of the 21$^{st}$ century. The source of this enhanced cold season runoff is a topic of ongoing research, but is likely connected to the increased frequency of landfalling ARs simulated in the CMIP5 models along the North American west coast (Warner et al., 2015; Gao et al., 2015; Payne and Magnusdottir, 2015; Radic et al., 2015; Warner et al., 2015; Warner and Mass, 2017; Curry et al., 2019). Under the projected warming, precipitation in the form of rainfall will be a key driver modulating Coast Mountains runoff intensity and frequency especially in the cold season. This may increase the risk of extreme flooding in the Coast Mountains and in the lower FRB, with associated implications for water management strategies in these areas.

The hydrologic response in mountainous regions varies considerably across the basin differentiating its flows mainly into snow-dominant or hybrid (rain and snow) regimes (Wade et al., 2001). The SP detection analysis suggests changes in the snow-dominant category arising more prominently across the Interior Plateau probably due to its lower mean elevation with smaller snowpack accumulation in winter and thus higher sensitivity to temperature increases during the cold season. Snowpack declines are most pronounced at temperatures near freezing that occur more often at Interior Plateau lower elevations during fall or spring. In higher mountainous regions with cold climates, flow regimes depend mainly on the moisture availability and elevation, with higher elevations having cooler air temperatures and thus longer periods of snow accumulation. Therefore the snowpack declines are less sensitive to temperature change in the Rocky Mountains. In the Coast Mountains, the flow regimes

will remain rain-dominant at lower elevations owing to abundant rainfall associated with the region's maritime climate.

This study deals solely with the impacts of projected changes in climate on the FRB's cold season runoff variability and flow regimes under strong greenhouse gas forcing on regional hydrology. Influences of other forcings, such as land cover change and glacier growth or loss, are not considered, similar to other recent modelling studies of projected climate change impacts in the FRB (Schnorbus et al., 2014, Shrestha et al., 2012 and Islam et al., 2017). The assumption of static land cover is probably inconsistent with the strong warming scenario (RCP8.5) used in this study, and represents a limitation of the approach that could be relaxed in future work. Several previous studies have shown that the sensitivity of runoff to forest cover change depends on a basin's size and regional characteristics (Wei et al., 2013; Zhang et al., 2017). These studies conclude that runoff response to forest cover change generally decreases with increasing basin size, with large snow-dominated basins being more resilient. Some support for this is also found in the study of Schnorbus et al. (2010), who utilized VIC simulations to quantify the impacts of idealized scenarios of mountain pine beetle and associated salvage harvesting across different watersheds within the FRB. They found that despite a large upstream sensitivity to land cover changes, the overall, integrated change in discharge at Hope, BC was quite low. Further, Havel et al. (2018) quantified the wildfire influence on streamflow in mountainous catchments and found small influence on cumulative runoff in the larger watershed. In addition, the version of the VIC model used in the present work does not simulate glacier dynamics and its contribution to runoff variability (although VIC does simulate large snow piles in specific cells that grow and ablate in response to climate forcings). Thus a realistic glacier model is required to address questions surrounding

the influence of anticipated declines in glacier mass on the interannual variability of runoff in the FRB. This is an ongoing effort of our research team focusing the development and application of an updated and re-engineered version of the VIC model (VIC-GL) that is coupled with snow mass and glacier dynamics models. Our future studies will utilize the VIC-GL model to quantify the contribution of the dynamic glaciers melt to runoff in several basins in western Canada including the FRB.

All of the historical CMIP5 driving data are essentially constrained to have the observed means and variability for the historical period in the statistical downscaling. The historical period variability therefore does not reflect large inter-model differences. Future variability does see the influence of inter-model differences and not just changes in variability due to forced climate change. The increase in the runoff variability may be therefore somewhat overestimated in our analysis. In addition to the downscaling methodology, the GCM dynamics, natural climate variability and hydrological modelling constitute additional sources of uncertainty in this study's projected hydrological changes. Furthermore, the hydrological model used in this study is integrated on a relatively coarse resolution (0.25°), which may not represent some aspects of the small-scale dynamics related to the FRB's complex topography. Nevertheless, considering the modest magnitude of the inter-model spread compared to the MME mean in the cold season, projected changes reported in this study are mainly induced by projected climate warming with increased precipitation variability. Such an increase in precipitation variability is also reported in Pendergrass et al. (2017) using CMIP5 MME precipitation without any downscaling method.

# 5 Conclusion

Climate change is expected to induce considerable hydroclimatic alterations in mid-latitude river basins around the globe. The results presented in this study provide supportive information on key hydrological changes under projected warming and changes in precipitation phase by focusing on the Fraser River Basin, a large mid-latitude basin with mountainous terrain and a strong maritime influence. In such basins, quantification of projected changes in the input precipitation and associated runoff changes is quite challenging due to complex mountainous topography and varying intensity of maritime influences. Using the CMIP5 projections, and a hydrological model, this study clearly demonstrates that in the warmer climate, changes in rain to snow ratio (Fig. 4) play a crucial role in modulating cold season snow accumulation (Fig. 5) and runoff (Fig. 6). The contribution of rainfall to overall increases in the mean runoff is quite evident in the coastal regions where the marine influences strengthen with time and hence increasing rainfall substantially. The increasing strength and frequency of maritime influences are possibly linked to the projected significant increase in atmospheric rivers as simulated by the driving CMIP5 GCMs. Using the cold season runoff from the three geoclimatic regions with cold season precipitation input to those regions, the multivariate linear regression analysis (Table 2) further confirms rainfall as a dominant contributor to runoff increases in the Coast Mountains. Overall, our results suggest that in the mid-latitude basins with maritime influences, the projected rainfall and runoff will rise much more rapidly. A larger fraction of the rainfall in these basins will become runoff producing rapidly rising cold season runoff.

Furthermore, our analysis of flow regime changes provides a new insight into expected regime transitioning in the snowfall-dominant basins with hybrid or rainfall-dominant flow regimes becoming

more prevalent. In the mountainous basins, such changes are strongly coupled to projected changes in precipitation and air temperature along with the varying range of elevations. Compared to high elevation mountainous regions, regime transitioning will be more apparent in the low elevations where near-freezing temperatures prevail. In snowmelt-dominant flow regimes with higher elevations, such

changes will most probably accelerate earlier onset of spring snowmelt and will decrease the magnitude of summer peak flow events.

While the results reported in this study are not precise representations of projected runoff changes due to several computational limitations, dataset uncertainties and modelling uncertainties such as those associated with land cover and glacier change, they nevertheless provide valuable insight to the

projected hydrological state of the mid-latitude mountainous basin and should be useful for planning and developing future water management resources including flood frequencies.

## Acknowledgments

This work was supported by the Natural Sciences and Engineering Research Council of Canada (NSERC) funded Canadian Sea Ice and Snow Evolution (CanSISE) Network, the Mountain Water

Futures (MWF) initiative of the Global Water Futures (GWF) program and the Nechako Environmental Enhancement Fund (NEEF). Thanks to colleagues from the Pacific Climate Impacts Consortium for their assistance and contributions in the VIC model implementation. The authors thank Michael Allchin (UNBC) for plotting Fig. 1a and for providing useful comments. The authors acknowledge the World Climate Research Programme's Working Group on Coupled Modelling, which is responsible for

CMIP5, and thank the climate modelling groups for producing and making available their GCM output.

For CMIP, the U.S. Department of Energy's Program for Climate Model Diagnosis and Intercomparison provides coordinating support and led development of software infrastructure in partnership with the Global Organization for Earth System Science Portals. Thank to Stephen Sobie (PCIC) for clarification of the BCCAQ2 method. Thanks to the anonymous reviewers and the handling editor (Dr. Chris DeBeer, University of Saskatchewan) for constructive comments and suggestions that greatly improved this paper.

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

**Table 1:** Characteristics of Water Survey of Canada (WSC) hydrometric stations and three geoclimatic regions within the FRB (Déry et al., 2012). The 30-year runoff mean and interannual variability (estimated by standard deviation) is calculated for each individual CMIP5-VIC simulation and all values are averaged to get the MME mean in the cold season. The inter-model spread in runoff mean and its interannual variability is indicated as uncertainty ranges (±) estimated by a 5-95% models range. The last column shows advances in the timing of the MME mean annual peak flow by 2080s.

| Basin (Abbreviation) [WSC ID] | Gauged area [km$^2$] | Latitude [°N], Longitude [°W] | Mean elevation [m] | Cold season mean and interannual variability of Runoff [mm yr$^{-1}$] | | | | | | Advance (days) by 2080s |
|---|---|---|---|---|---|---|---|---|---|---|
| | | | | 1990s (1980-2009) | | 2050s (2040-2069) | | 2080s (2070-2099) | | |
| | | | | Mean | Variability | Mean | Variability | Mean | Variability | |
| Rocky Mountains | - | - | 1567 | 160 ± 3 | 25 ± 2 | 205 ± 12 | 51 ± 7 | 270 ± 22 | 65 ± 9 | - |
| Interior Plateau | - | - | 1101 | 58 ± 0 | 10 ± 1 | 78 ± 4 | 19 ± 2 | 102 ± 7 | 22 ± 2 | - |
| Coast Mountains | - | - | 1296 | 450 ± 10 | 90 ± 7 | 606 ± 23 | 139 ± 11 | 730 ± 34 | 158 ± 15 | - |
| Upper Fraser (UF) [08KB001] | 32400 | 54.01, 122.62 | 1308 | 76 ± 8 | 29 ± 4 | 107 ± 13 | 46 ± 7 | 133 ± 18 | 54 ± 9 | 20 |
| Quesnel (QU) [08KH006] | 11500 | 52.84, 122.22 | 1173 | 119 ± 3 | 24 ± 2 | 142 ± 8 | 42 ± 6 | 182 ± 16 | 52 ± 7 | 18 |
| Thompson-Nicola (TN) [08LF051] | 54900 | 50.35, 121.39 | 1747 | 47 ± 2 | 13 ± 1 | 74 ± 7 | 28 ± 5 | 113 ± 12 | 38 ± 4 | 25 |
| Chilko (CH) [08MA001] | 6940 | 52.07, 123.54 | 1756 | 95 ± 2 | 20 ± 2 | 119 ± 6 | 34 ± 5 | 151 ± 10 | 42 ± 4 | 35 |
| Fraser at Hope (LF) [08MF005] | 217000 | 49.38, 121.45 | 1330 | 83 ± 2 | 13 ±1 | 109 ± 6 | 26 ± 4 | 142 ± 10 | 33 ±4 | 25 |

**Table 2:** Decomposition of the key drivers affecting cold season runoff changes in the 2080s. Contributions are in % estimated using the multivariate linear regression (MLR, described in section 2.4.2) model using monthly time series from 21 CMIP5 simulations. $R^2$ provides the variance explained by all three variables. The gridcell averaged contributions are estimated only for statistically significant values at p < 0.05. Change in each variable is normalized by the total runoff change to estimate its contribution. The contributions listed in last three columns do not necessarily sum exactly 100% due to rounding off and/or masking of insignificant gridcells before taking area-averages.

| Region | Runoff Change 2080s (%) | | $R^2$ | Contribution to Runoff Change (%) | | |
|---|---|---|---|---|---|---|
| | VIC | MLR | | Rainfall | Snowfall | Mean Air Temperature |
| Coast Mountains | 121 | 146 | 0.70 | 61 | 8 | 32 |
| Interior Plateau | 114 | 140 | 0.42 | 44 | 4 | 46 |
| Rocky Mountains | 79 | 117 | 0.45 | 44 | 6 | 50 |
| Fraser River Basin | 107 | 133 | 0.45 | 47 | 8 | 46 |

**Table 3:** CMIP5-VIC simulated projected change in snowmelt-dominant regimes. Values are in % calculated using the ratio of the sum of SPs years and the total number of years over all 21 CMIP5-VIC simulations. The uncertainty ranges (±) indicate inter-model spread in the MME mean values indicated by a 5-95% models range.

| Region | Snowmelt-Dominant Regime (%) | | |
|---|---|---|---|
| | **1990s (1980-2009)** | **2050s (2040-2069)** | **2080s (2070-2099)** |
| Coast Mountains | 95±1 | 74±3 | 57±4 |
| Interior Plateau | 100±0 | 73±5 | 43±7 |
| Rocky Mountains | 100±0 | 88±3 | 64±6 |
| UF | 95±9 | 95±9 | 93±9 |
| QU | 100±0 | 98±1 | 85±5 |
| TN | 100±0 | 98±1 | 80±8 |
| CH | 100±0 | 100±0 | 95±1 |
| LF | 100±0 | 98±1 | 81±7 |

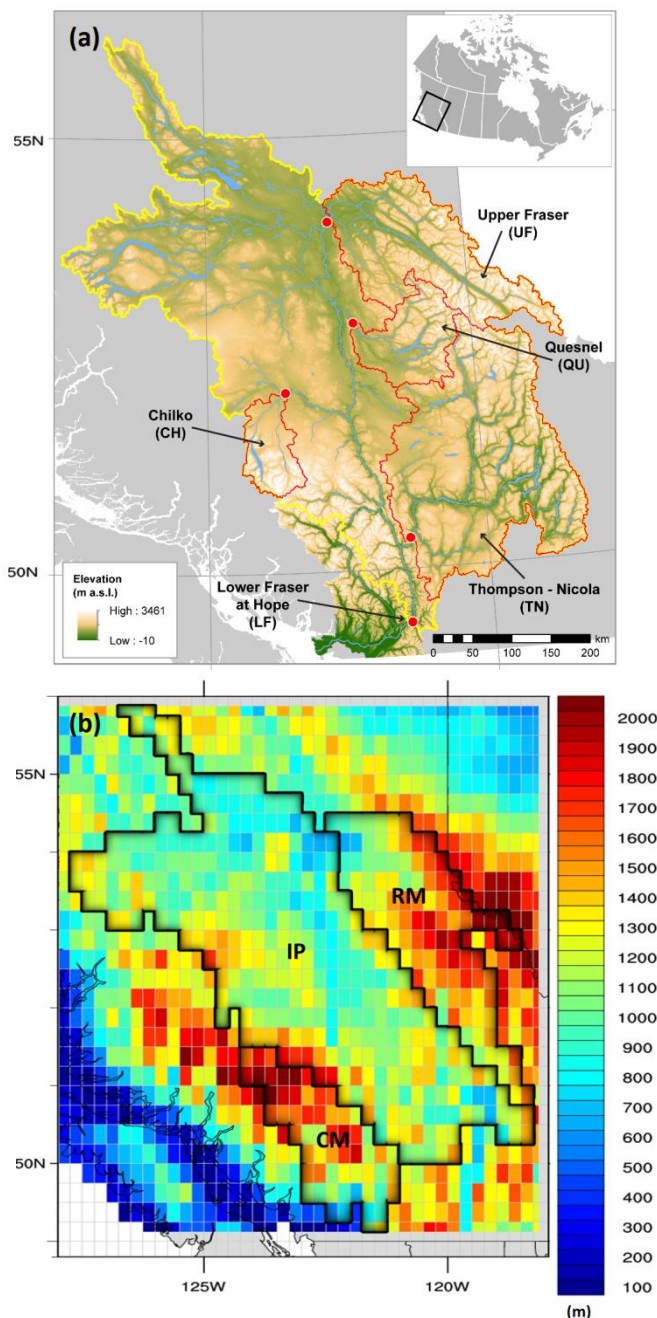

**Figure 1:** (a) Digital elevation model of the FRB with identification of sub-basins: Upper Fraser (UF), Quesnel (QU), Chilko (CH), Thompson–Nicola (TN), and Fraser at Hope (LF). (b) Elevation map highlighting the Fraser River Basin and three geoclimatic regions. RM, IP and CM represent the Rocky Mountains, Interior Plateau and Coast Mountains, respectively.
5 Elevations are in meters and the horizontal grid resolution is that of the VIC hydrological model (0.25°).

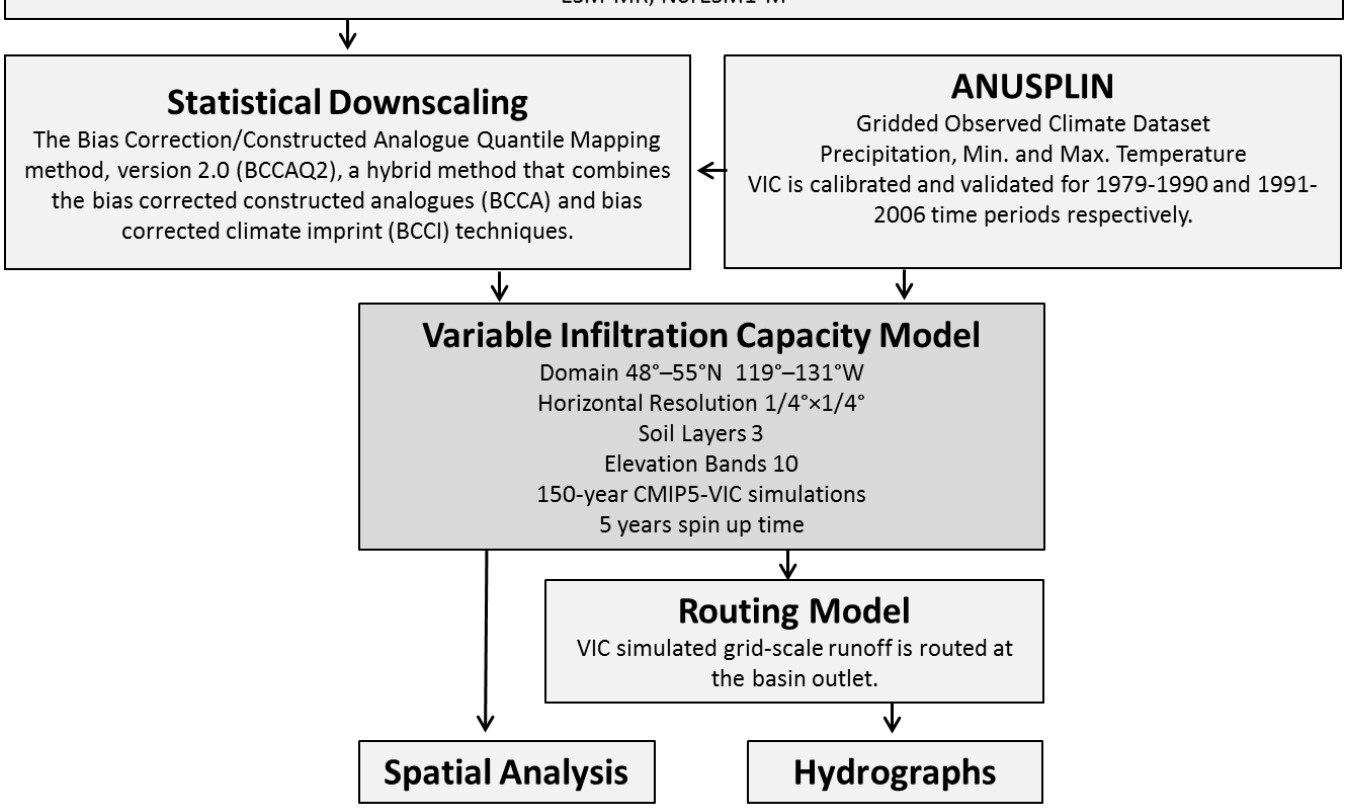

**Figure 2:** Block diagram of the VIC model experimental setup and analysis.

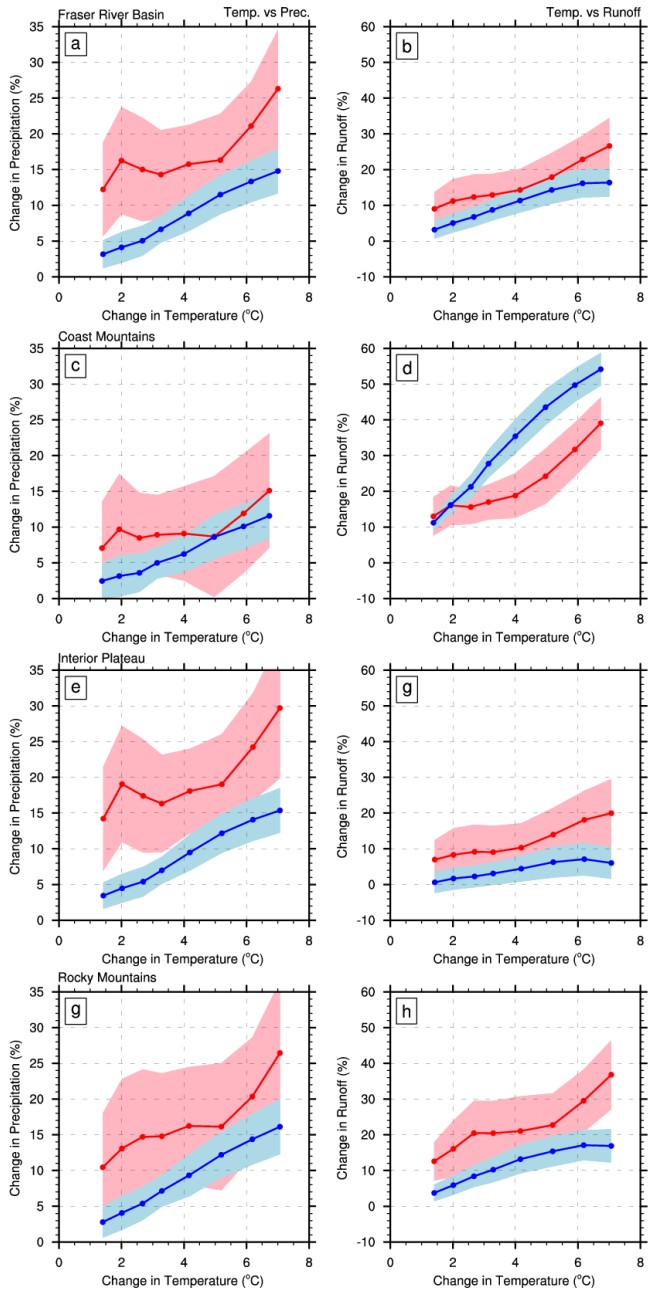

**Figure 3:** Rate of change of mean and interannual variability with warming for precipitation (a, c, e, g) and runoff (b, d, f, h) during the cold season. Changes in the cold season mean (blue line) and interannual standard deviation (red line) are shown as a function cold season mean temperature change for the Fraser River basin (a, b), Coast Mountains (c, d), Interior Plateau (e, f) and Rocky Mountains (g, h). Each marker indicates a 30-year period centred on consecutive decades between 2010 and 2080 relative to the 1990s base period. Shading represents inter-model spread in 30-year means as indicated by a 5-95% models range.

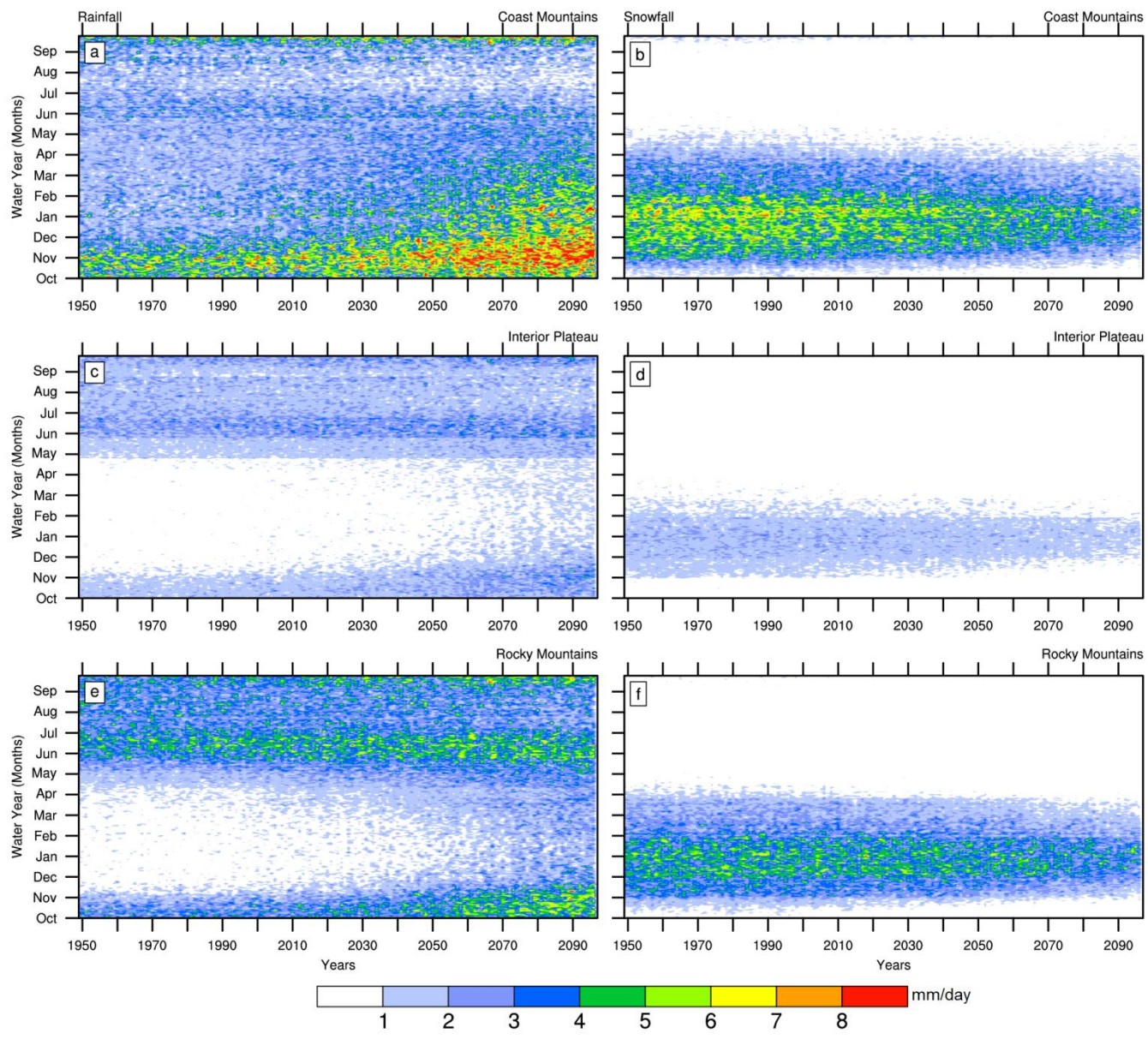

**Figure 4:** Partitioning of CMIP5 models MME mean total precipitation into daily rainfall (a, c, e) and snowfall (b, d, f) for the Coast Mountains (a, b), Interior Plateau (c, d) and Rocky Mountains (e, f). Values are regional spatial averages over the three geoclimatic regions. Units are mm day$^{-1}$.

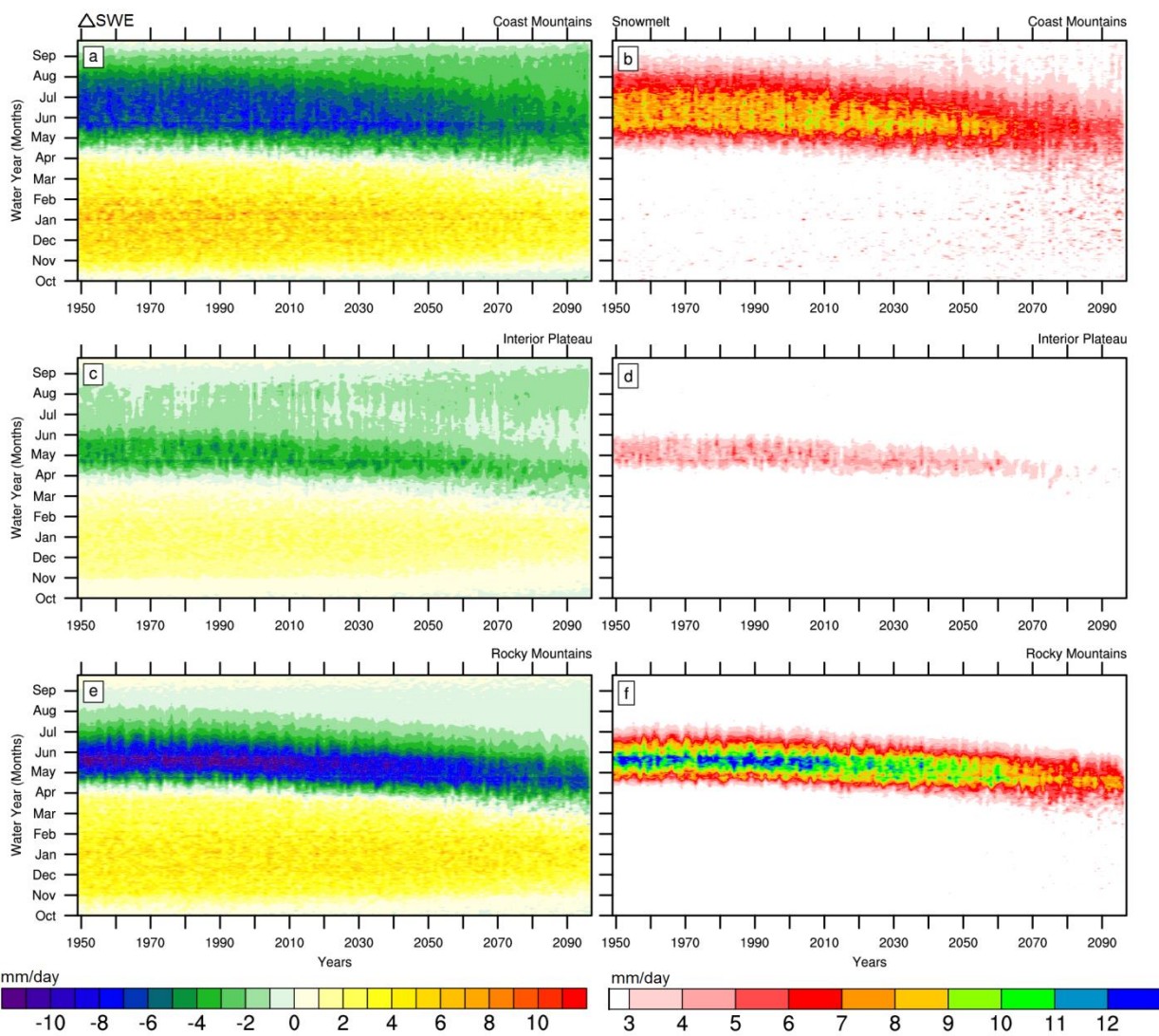

**Figure 5:** CMIP5-VIC simulated MME mean ΔSWE (daily SWE rate) (a, c, e) and snowmelt (b, d, f) for the Coast Mountains (a, b), Interior Plateau (c, d) and Rocky Mountains (e, f). In panels (a), (c) and (e), values greater than 0 represent snow accumulation while those below 0 indicate snow ablation. Units are mm day$^{-1}$.

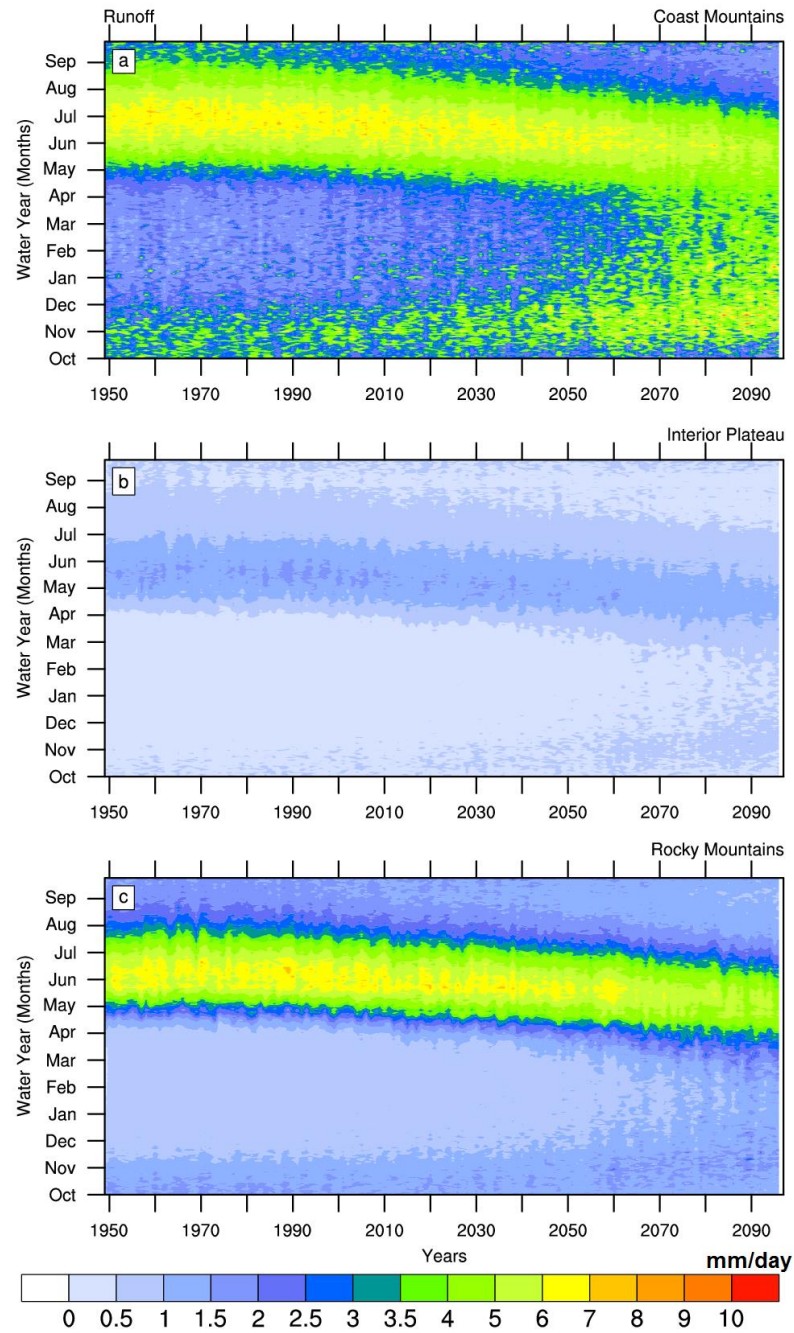

**Figure 6:** CMIP5-VIC simulated daily mean runoff for the Coast Mountains (a), Interior Plateau (b) and Rocky Mountains (c). Values are spatial averages over geoclimatic regions. Units are mm day[-1]. Runoff units are an equivalent regional average rainfall rate rather than a discharge rate.

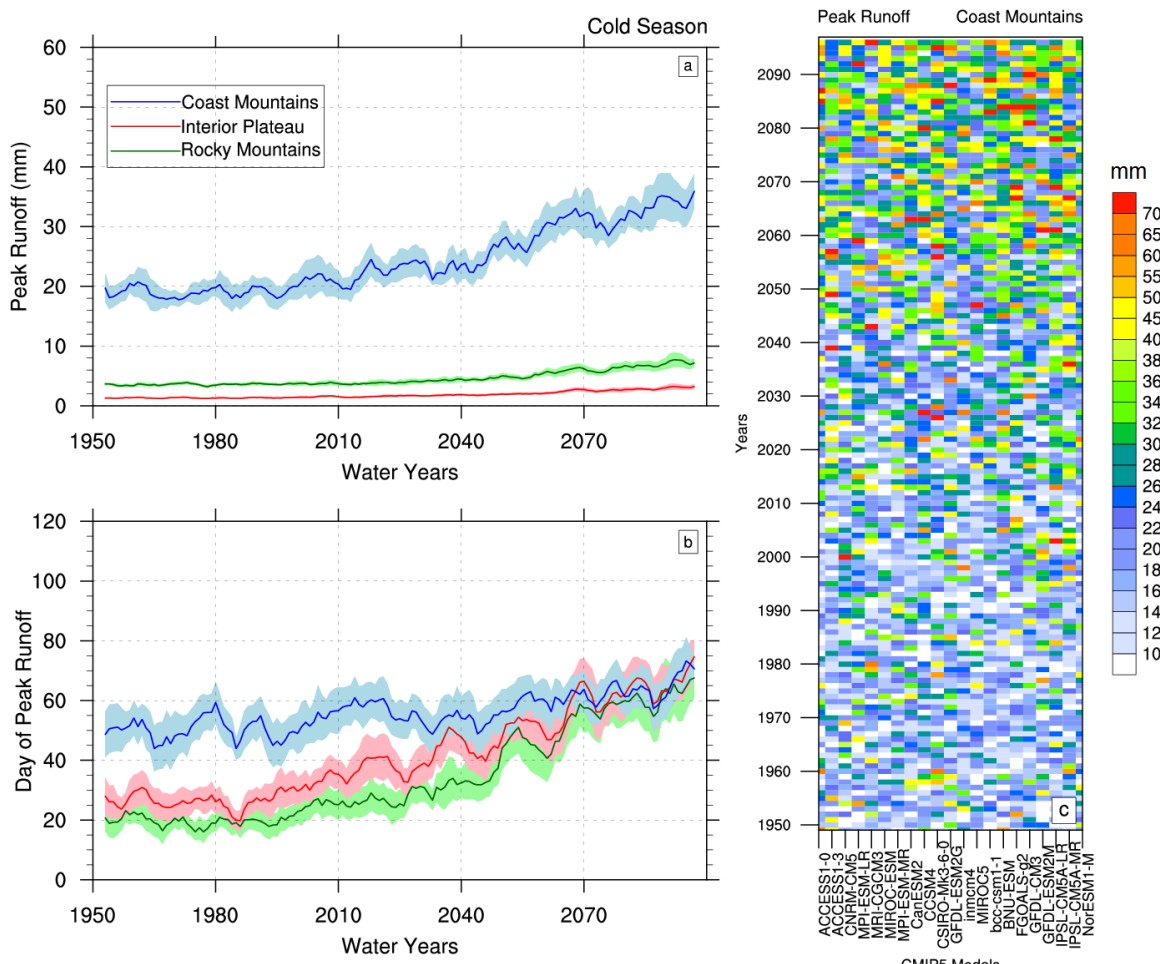

**Figure 7:** CMIP5-VIC simulated peak runoff (a) and corresponding day of the peak runoff (b) in the cold season of the water year for Coast Mountains, Interior Plateau and Rocky Mountains. Solid curves are for the MME mean and shading represents inter-model spread as represented by a 5-95% models range. A 5-year running mean is applied to smooth variations in all curves. Y-axis values in panel (b) represent days of water year where 0 corresponds to 1 October. Panel (c) displays cold season peak runoff simulated by individual CMIP5-VIC simulations for the Coast Mountains. Units are mm.

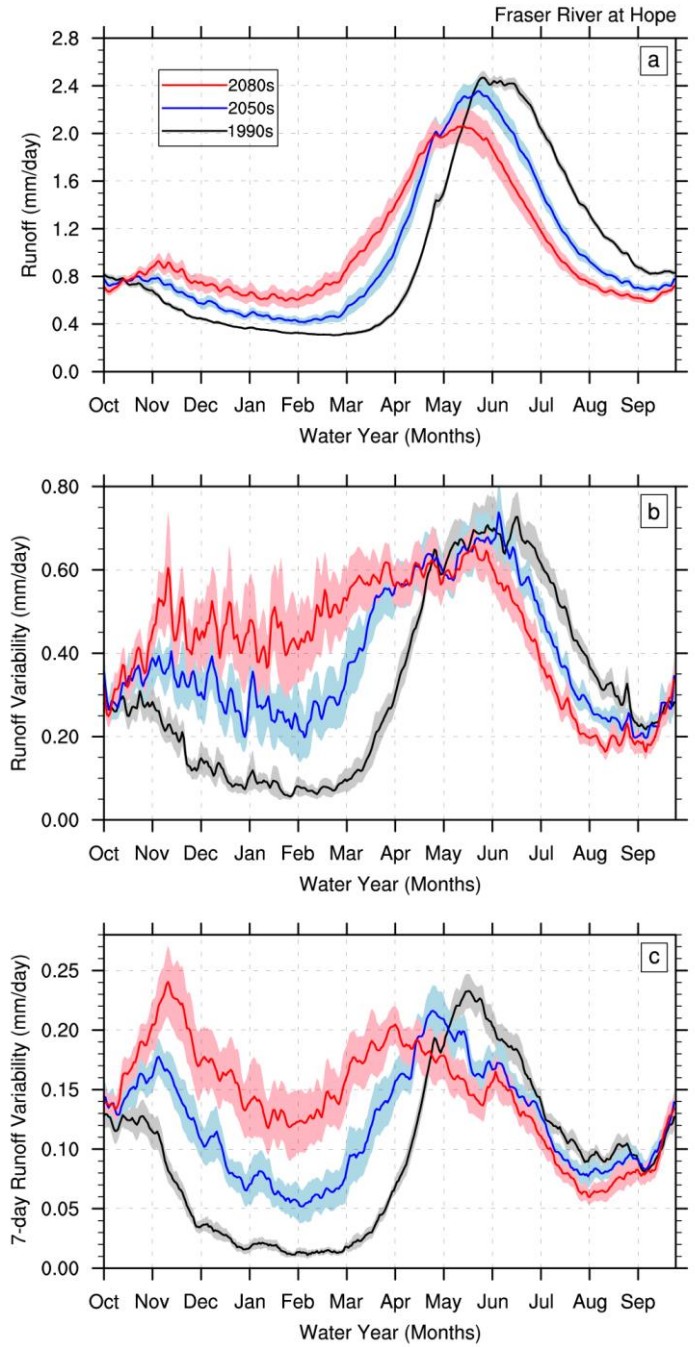

**Figure 8:** CMIP5-VIC simulated daily runoff (normalized discharge) mean (a), variability (b) and 7-day variability (c) for the Fraser River at Hope. Black, blue and red curves represent the MME mean for the 1990s, 2050s and 2080s, respectively. Shading represents inter-model spread as indicated by a 5-95% models range.

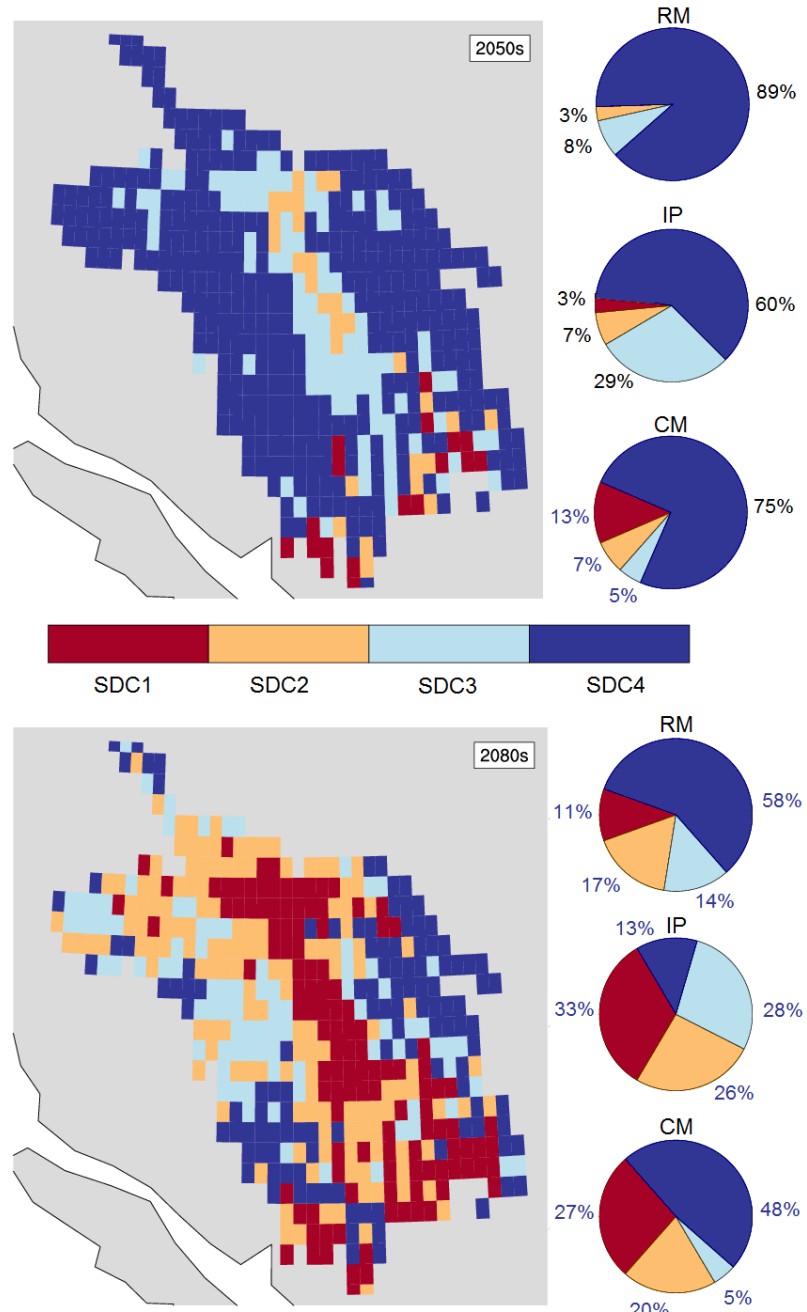

**Figure 9:** CMIP5-VIC simulated MME mean projected snowmelt-dominant Categories (SDC) in the 2050s and 2080s. SDCs are classified using the SP detection algorithm (described in section 2.5) and are based on changes (%) in the fraction of years with SP. Pie charts show corresponding change in fraction of gridcells (%) in each SDC for the Rocky Mountains (RM), Interior Plateau (IP) and Coast Mountains (CM) geoclimatic regions.

