# Peer review of "Quantifying projected changes in runoff variability and flow regimes of the Fraser River Basin, British Columbia"

_Hydrology and Earth System Sciences, 2018_

## Referee Comment (RC1) · Anonymous Referee #1 · 2 Jun 2018

In this study, the authors examined the influence of future climate scenarios on streamflow in the Fraser River basin in British Columbia, Canada. They used statistically downscaled output from 21 GCMs for the RCP 8.5 emissions scenario, using one realization from each GCM. The authors used the VIC hydrologic model, which has been applied in previous studies to look at the effects of climate and land-cover change on streamflow. Key results are that the basin will transition from a snow-dominated regime to a more rain-dominated regime, and that flow variability will increase in winter, with an increase in the magnitude of cold-season peak flows.

Overall, the study appears to have been conducted in a competent manner using up-

to-date approaches for generating the future climate scenarios. I expect that the results will be of great interest to the agencies involved in managing water-related resources and hazards in the Fraser basin. However, the manuscript reads like a regional case study, and I struggled to discern how this work contributes novel and significant knowledge in the context of the international readership of HESS. The shift from snow- to rain-dominated regimes in mountainous mid-latitude catchments has been identified in dozens, if not hundreds, of earlier climate-impact studies published in the international literature.

Based on descriptions of the model set-up in earlier work by the authors, I infer that land cover was held constant through the simulations. In reality, however, land-cover will evolve, particularly in response to widespread forest disturbance related to the Mountain Pine Beetle outbreak that began in the 1990s, and the salvage logging that followed. In addition, glacier retreat will undoubtedly influence the hydrology of some of the mountainous headwaters. An important question is the extent to which these land-cover changes would amplify or diminish the effects of climatic change.

On balance, I am not fundamentally opposed to the publication of this work, but I believe the authors need to make a more convincing case that this study represents an internationally significant contribution to the literature and is not just a regional case study. The authors need to highlight what is novel about this work when considered within the broader context of the international literature. I should note that I have not kept up with the climate-impacts literature for a few years, and I may not have the background to appreciate the novelty of this work without it being spelled out more explicitly.

---

## Referee Comment (RC2) · Anonymous Referee #2 · 9 Jun 2018

This is a generally well-written paper that discusses changes in runoff variability and flow regimes in the Frasier River Basin under climate change. The authors analyze 21 downscaled CMIP5 simulations that have been used as input to a VIC model implementation at 0.25° resolution. While the paper is generally well-written, the study itself is mostly routine and is not sufficiently novel in its current form that I can recommend publication in HESS.

General comments:

1. The paper is purely descriptive in its analysis. The authors describe the results from the model simulations, but make no real attempt to analyze and interpret them. For

example, which specific processes contribute most to the increase in runoff variability? How does this increased variability affect the salmon population (the authors repeatedly make the point that the Frasier supports the largest migration of Pacific Ocean sockeye salmon in the world)?

2. As is, the findings mostly have regional interest, as there is no new methodological development nor are any of the findings particularly surprising. Warming in climates with a large seasonal snow component will lead to larger flows in winter because of a shift from snowfall to rain, mid-season melt, and earlier melt, but this has been widely reported for similar basins in western North America and Europe. The field has progressed to where this may be extremely useful information for local water managers, but the study design and findings by themselves are not sufficient for publication in a scientific journal.

3. The paper is not significantly different from an earlier paper by the same lead author in Journal of Hydrometeorology (doi:10.1175/JHM-D-16-0012.1), which discusses the same modeling chain and setup (some of the figures are near identical and should be attributed at the very least). That paper is based on a smaller model ensemble and focuses more on changes in the mean climate / hydrology rather than changes in variability. If the authors choose to focus on variability in this paper, then I would encourage them to analyze what this increase in variability actually means for the basin.

4. The manuscript lacks a clear conclusion section as the authors have a single combined discussion and conclusion without a clear take-home message. I would suggest splitting these components as it emphasizes the need to have a clear conclusion that adds to the existing body of knowledge and that is focused on findings that are of wider interest than the local changes in the Frasier River basin.

Specific comments:

a. p.5 l.3-4: While the authors state that it "[...] is important to evaluate such regime transitions on regional scales while characterizing snowmelt and rainfall driven flows

independently", they never clearly state why this is important and how they will use this analysis.

b. Section 2: This section should reference their earlier work (Islam et al., 2017) more directly, as much of the model setup is the same, in particular the setup of the hydrological model. As is, the section is rather uneven. It goes into great detail regarding the resolution of the ANUSPLIN dataset ("having a spatial resolution of about 9.26 km in the meridional direction and one that varies proportionally to the cosine of latitude in the zonal direction.") but says nothing about the VIC calibration or setup. Incidentally - is the ANUSPLIN dataset simply a 5 arcmin resolution (1/12°)?

c. Section 2.2: In addition to the strengths, the authors should also address the shortcomings of the downscaling techniques that they use, especially since they look at variability in daily time series. For example, Gutmann et al. (2014) noted that BCCA overestimates wet day fraction and underestimates extreme events. Perhaps the combination with BCCI fixes this, but that would be good to discuss.

d. p.9 l.1: Wu et al. (2011) does not describe a routing scheme, but simply provides routing networks at different spatial resolutions. From the sentence that follows it appears that the authors have used the Lohmann routing scheme. This should be clarified.

e. Section 2.3: The authors do not provide sufficient detail about the VIC setup. It is fine to refer to their earlier paper, but it would be good to mention model resolution, a two-line summary of the source of the parameters, etc. That would be more useful than the long list of references to previous uses of the VIC model (p. 9 second paragraph).

f. p.11 l.1-2: "Peak runoff during the cold season was computed between 1 October and 1 March when the 3-day running mean daily air temperature exceeds 0°C at each gridcell." Why the extra condition based on air temperature?

g. Section 2.4.2: This section needs to be streamlined. The equations are unnecessary, since most of us know how to calculate a mean and variance for a data set.

h. In the results section I found the narrative hard to follow in part because of the way in which the authors use abbreviations to refer to the different sub-basins. Sentences such as "The advance in the timing of the annual peak flow in these sub-basins is slightly less than for the FRB as a whole (∼20 days for UF, ∼18 days for QU, ∼25 days for TN and ∼35 days for CH) [...]" are difficult to read. The numbers may be more effectively presented in a Table, which allows the text to focus on some particular insight that can be derived from this.

i. Figures were generally of good quality.

References:

Gutmann, E., T. Pruitt, M. P. Clark, L. Brekke, J. R. Arnold, D. A. Raff, and R. M. Rasmussen, 2014: An intercomparison of statistical downscaling methods used for water resource assessments in the United States. Water Resour Res, 50, 7167-7186, 10.1002/2014wr015559.

―――――――――――――――――

---

## Author Comment (AC1) · 6 Jul 2018

**Responses to Reviewers**

*We are thankful to anonymous Reviewer #1 and #2 for their comprehensive comments on our manuscript. Here we provide a general overview of our responses (in bold italic) to the major comments submitted by each referee.*

**Reviewer # 1:**

In this study, the authors examined the influence of future climate scenarios on streamflow in the Fraser River basin in British Columbia, Canada. They used statistically downscaled output from 21 GCMs for the RCP 8.5 emissions scenario, using one realization from each GCM. The authors used the VIC hydrologic model, which has been applied in previous studies to look at the effects of climate and land-cover change on streamflow. Key results are that the basin will transition from a snow-dominated regime to a more rain-dominated regime, and that flow variability will increase in winter, with an increase in the magnitude of cold-season peak flows.

*Thank you kindly for reviewing this manuscript.*

**1.** Overall, the study appears to have been conducted in a competent manner using up to-date approaches for generating the future climate scenarios. I expect that the results will be of great interest to the agencies involved in managing water-related resources and hazards in the Fraser basin. However, the manuscript reads like a regional case study, and I struggled to discern how this work contributes novel and significant knowledge in the context of the international readership of HESS.

*1. We appreciate the careful review of our manuscript, and thank the Reviewer for his/her perceptive comments. While a majority of HESS papers are in fact regionally focused, we agree that we have not paid sufficient attention to what this particular "regional case study" can teach us about other similar regions in the world. In fact, as we discuss below, the somewhat unusual physical setting of the FRB can teach us a great deal about hydroclimatic change in a mid-latitude mountainous basin with strong maritime influences. Furthermore, as our manuscript is targeted for the HESS special issue on "Understanding and prediction earth system and hydrological changes in cold regions," in our revised manuscript we better distinguish projected changes in the FRB that are fairly universal from those that are case-specific.*

2. The shift from snow-to rain-dominated regimes in mountainous mid-latitude catchments has been identified in dozens, if not hundreds, of earlier climate-impact studies published in the international literature.

*2. We agree with the Reviewer that there are many different studies reporting snow- to rain-dominated regime changes. However, the FRB does in fact differ in important ways from other mountainous mid-latitude catchments. While it is indeed a mid-latitude nival basin, it extends from the Pacific coast to the continental interior, meaning that it is also maritime-influenced. Specifically, the hydrologic response to warming in the FRB is influenced by two possibly confounding factors: first, the change of phase from snow to rain; and second, the very significant increase in atmospheric moisture supply (atmospheric rivers) to the North*

*American west coast as projected in the CMIP5 future projections (Payne & Magnusdottir, 2015; Radic et al., 2015; Warner et al., 2015; Warner & Mass, 2017). It is the latter feature that is somewhat unusual compared to other large, mid-latitude mountainous basins worldwide. For example, while small, mountainous catchments on the Norwegian coast are also strongly influenced by atmospheric rivers, they are also characterized as hybrid pluvial-nival regimes at present, and lack the extensive interior snowpack that exists in the FRB headwaters of the Canadian Rockies. An important research question that we aim to address is whether this interior snowpack will increase or decrease in response to these large-scale changes, because the exact, geographic and seasonal change in moisture supply needs to be included as part of the modelling chain. We feel that these are questions that can only be addressed in the region we selected using our carefully designed modelling framework. In the revised manuscript and Supplemental Material, we have now included results that make the "added value" of our simulation strategy for answering these types of questions much more evident.*

3. Based on descriptions of the model set-up in earlier work by the authors, I infer that land cover was held constant through the simulations. In reality, however, land-cover will evolve, particularly in response to widespread forest disturbance related to the Mountain Pine Beetle outbreak that began in the 1990s, and the salvage logging that followed. In addition, glacier retreat will undoubtedly influence the hydrology of some of the mountainous headwaters. An important question is the extent to which these land-cover changes would amplify or diminish the effects of climatic change.

*3. While an investigation of land cover effects might be of interest, it is beyond the scope of the present effort, which deals solely with the impacts of projected changes in climate on the FRB's cold season runoff variability and flow regimes under strong greenhouse gas forcing. We point out that any choice of land use / land cover scenario is arbitrary considering that all future changes are conditional on a given scenario. In addition, it would be difficult to predict how forest composition and disturbance regimes will change in the future. We also suspect that the effect of glacier losses and ultimately the end of glacier wasting will not have a fundamental impact on future flow regimes at the scale of the FRB. In fact the hydrological model used in our study does store and ablate frozen water at high elevations in the form of piles of snow that grow over time under historical climate conditions. While those snow piles do not have quite the same surface properties as ice (e.g., they do not flow), they represent crude glaciers in model simulations that grow during the historical period in some locations, and subsequently ablate as melting outpaces deposition.*

*Most importantly, it is likely that both land cover change and glacier retreat constitute a second-order forcing compared to the dominant effect of the strong increase in greenhouse gas forcing under the RCP8.5 scenario. Once the primary hydrologic response to RCP8.5 has been estimated, a follow-up study focused on the effects of land cover change and glacier retreat (with a hydrologic model including dynamic glaciers) might be merited.*

4. On balance, I am not fundamentally opposed to the publication of this work, but I believe the authors need to make a more convincing case that this study represents an internationally significant contribution to the literature and is not just a regional case study. The authors need to highlight what is novel about this work when considered within the broader context of the

international literature. I should note that I have not kept up with the climate-impacts literature for a few years, and I may not have the background to appreciate the novelty of this work without it being spelled out more explicitly.

*4. We thank the Reviewer for this suggestion and reiterate that our revision will better highlight the novel aspects of this work. In our revised Results and Discussion sections, we have provided a more fulsome analysis of future-projected hydrological changes in the FRB (described in detail in our response to point 1 of Reviewer 2) that clearly sets our work apart from that on other mountainous mid-latitude catchments in the published literature.*

**Reviewer # 2:**

This is a generally well-written paper that discusses changes in runoff variability and flow regimes in the Frasier River Basin under climate change. The authors analyze 21 downscaled CMIP5 simulations that have been used as input to a VIC model implementation at 0.25 resolution. While the paper is generally well-written, the study itself is mostly routine and is not sufficiently novel in its current form that I can recommend publication in HESS.

*We thank the Reviewer for providing insightful comments and suggestions on our manuscript. The key issue touched on here regarding the novelty of our work is addressed in detail under point 2 below.*

1. The paper is purely descriptive in its analysis. The authors describe the results from the model simulations, but make no real attempt to analyze and interpret them. For example, which specific processes contribute most to the increase in runoff variability? How does this increased variability affect the salmon population (the authors repeatedly make the point that the Frasier supports the largest migration of Pacific Ocean sockeye salmon in the world)?

*1. Thank you for raising this issue. We agree with the referee's concerns and have placed additional emphasis on the analysis and explanation of model results throughout the Results and Discussion sections. In addition, we performed additional analysis to address the Reviewer's question, "which specific processes contribute most to the increase in runoff variability?" First, we examined the rate of change of runoff variability and mean with respect to the amount of warming in the Coast, Interior Plateau and Rocky Mountains subregions of the FRB. In the Coast Mountains, we found that the change in cold season mean runoff is significantly larger than the change in runoff variability, unlike in the other two regions. This finding helps to explain why the signal of increased future runoff is so much more evident in the Coast Mountains region. Second, we employed a multivariate linear regression model to decompose the cold season runoff monthly variability into separate contributions from precipitation and temperature. This procedure allows us to determine the contribution of each key driver to the simulated runoff variability at individual gridcells. Furthermore, the multilinear regression model explains 70-90% of the total variance in runoff over a large portion of the basin.*

*The question of impacts on the salmon population is beyond the scope of this study considering that such an impact assessment would involve an examination of water quantity, temperature, oxygen content, and also a solid knowledge of salmonid biology which is not our area of expertise. However, in our revised Discussion section, we do mention further possible links between hydrological changes in the FRB and salmon migration.*

2. As is, the findings mostly have regional interest, as there is no new methodological development nor are any of the findings particularly surprising. Warming in climates with a large seasonal snow component will lead to larger flows in winter because of a shift from snowfall to rain, mid-season melt, and earlier melt, but this has been widely reported for similar basins in western North America and Europe. The field has progressed to where this may be extremely useful information for local water managers, but the study design and findings by themselves are not sufficient for publication in a scientific journal.

*2. We thank the Reviewer for raising these points, which are similar to those raised by Reviewer #1. With regard to the comment, "the findings mostly have regional interest," we point out that, in fact, most papers in published in HESS have a regional focus. However, we agree that we did not pay sufficient attention to what our study of the FRB could teach us about other similar regions around the world. As mentioned in the response to Reviewer #1's point 2, we agree that we could better highlight the novel aspects of our work, and have made several changes to the revised manuscript to do just that. Previous studies that examined future hydroclimatic changes in the FRB were mostly focused on monthly and annual time scale differences in mean climatology and hydrographs. By contrast, there has been relatively little work quantifying cold season, daily time scale flow variability and regime transitions in the FRB. These research goals necessitate the use of a large model ensemble, an effective downscaling and bias-correction method (we used BCCAQ2 due to its proven utility at the daily time scale,) and a robust snowmelt detection algorithm. Our determination of snowmelt-dominant categories (SDCs) and their future change, carried out at fine spatial scale in Section 3.3, was not mentioned by either Reviewer. Yet, this is to our knowledge an original contribution in a hydroclimatic modelling context for any basin worldwide to study projected runoff regime transition. This study is therefore not a routine effort and represents a significant advance over what has appeared in the published literature. Nevertheless, we are strongly motivated by the Reviewer's comment to better emphasize the novel methodology and key research results of this study.*

*Regarding the Reviewer's comment that, "Warming in climates with a large seasonal snow component will lead to larger flows in winter because of a shift from snowfall to rain, mid-season melt, and earlier melt, …", we agree that most prior research bears this out. However, the situation is not so simple in the FRB, since there is evidence of a significant increase in atmospheric rivers impacting the North American west coast as projected in the CMIP5 models. This makes the FRB somewhat unusual compared to other mid-latitude mountainous basins isolated from maritime influences, and is an important reason why we chose this basin for our study, as discussed in more detail under Reviewer 1, point 2. In the revised manuscript and Supplemental Material, we have now included results (among them the change in the future SWE and snowcover distribution and their dependence on elevations) that make the added value of our simulation strategy for answering these types of questions much more evident.*

3. The paper is not significantly different from an earlier paper by the same lead author in Journal of Hydrometeorology (doi:10.1175/JHM-D-16-0012.1), which discusses the same modeling chain and setup (some of the figures are near identical and should be attributed at the very least). That paper is based on a smaller model ensemble and focuses more on changes in the mean climate / hydrology rather than changes in variability. If the authors choose to focus on variability in this paper, then I would encourage them to analyze what this increase in variability actually means for the basin.

*3. In fact, as mentioned above, both the methodology and the research focus of the present work differ significantly from the earlier study of Islam et al. (2017), although we acknowledge that these aspects were not sufficiently emphasized in the submitted manuscript. In the Methods section of the revised paper, we highlight the methodological improvements of BCCAQ2 downscaling and bias correction over that of BCSD, along with the utility of the*

*snowmelt detection algorithm we employ. Specifically, we emphasize that future-projected changes in daily flow variability and runoff extremes can only be examined using the chosen downscaling method, not BCSD. We also point out that Islam et al. (2017) examined projections only out to the 2050s using 12 CMIP5 models, while we use a 21-model CMIP5 ensemble and provide projections to 2100. The use of 21 models in this study allows us to better sample the uncertainty in driving GCMs. Indeed, the use of BCCAQ2 and the extended time horizon lead to new insights into projected changes in regional runoff and its variability that are not previously available in literature (e.g., the strongly increasing peak runoff in the Coast Mountains after ~2040). Finally, in our revised Discussion, we investigate the implications of the simulated increase in daily time scale variability in the FRB.*

4. The manuscript lacks a clear conclusion section as the authors have a single combined discussion and conclusion without a clear take-home message. I would suggest splitting these components as it emphasizes the need to have a clear conclusion that adds to the existing body of knowledge and that is focused on findings that are of wider interest than the local changes in the Frasier River basin.

*4. In our submitted manuscript, a separate Conclusions section has been created after the Discussion, to provide readers with a succinct and clear take home message.*

Specific comments:

*All specific comments will be addressed and incorporated in our revised manuscript.*

*References:*

*Payne, A. E. and Magnusdottir, G.: An evaluation of atmospheric rivers over the North Pacific in CMIP5 and their response to warming under RCP 8.5, J. Geophys. Res., 120(21), 11,173-11,190, doi:10.1002/2015JD023586., 2015.*

*Radic, V., Cannon, A. J., Menounos, B. and Gi, N.: Future changes in autumn atmospheric river eventsin British Columbia, Canada, as projectedby CMIP5 global climate models, J. Geophys. Res. Atmos., 120, 9279–9302, doi:10.1002/2015JD023279.Received, 2015.*

*Warner, M. D., Mass, C. F. and Salathé, E. P.: Changes in Winter Atmospheric Rivers along the North American West Coast in CMIP5 Climate Models, J. Hydrometeorol., 16(1), 118–128, doi:10.1175/JHM-D-14-0080.1, 2015.*

*Warner, M. D. and Mass, C. F.: Changes in the Climatology, Structure, and Seasonality of Northeast Pacific Atmospheric Rivers in CMIP5 Climate Simulations, J. Hydrometeorol., 18(8), 2131–2141, doi:10.1175/JHM-D-16-0200.1, 2017.*

---

## Author Comment (AC2) · 6 Jul 2018

Responses to reviewer's comments are provided in the attached document.

Please also note the supplement to this comment:
https://www.hydrol-earth-syst-sci-discuss.net/hess-2018-232/hess-2018-232-AC2-supplement.pdf
* * *

---

## Referee Comment (RC3) · Anonymous Referee #3 · 9 Jul 2018

My first assessment was similar to that of the previous reviewers: what has been modelled for the Fraser River has been modelled and reported many times before: changes in mean flow, regime, snow-rain ratio, etc. Abstract and conclusion provide little new information and the international reader doesn't know what knowledge gain to transfer to other regions. In this context we should remember that HESS has the same requirements for special issue papers as for regular contributions. Manuscripts submitted as type 'research articles' should 'clearly advance our understanding', ms type 'cutting-edge-case study' needs to provide all data to serve others as testbed e.g. for models (from the HESS website). The current manuscript is perhaps in-between. A symptomatic indicator is the start of Section 5 "..overall question...how...precipitation phase

and variability will modulate the FRB's runoff variability and flow regimes". Instead of this case study view, the science question should be how cold climate hydrology transitions to temperate climate hydrology - the FRM just happens to be considered the case that is used for illustration.

However, with the running model at hand and gauging from the responses given already there is potential to focus on a particular process or phenomenon that is not yet well understood and is still specific to cold regions transitioning to temperate climate. Some of the analyses on the variability and pulses etc. that are presented here stand out and may provide a nice starting point. They are the ones that could be made the sole focus, analysed more specifically and quantitatively to make this an original contribution specifically dealing with features of the transition from seasonal snow to more rainfall-runoff dominated flow dynamics. It would have been very interesting, for example, to see the analysis on the daily to weekly variability expanded more systematically to scale and quantities - e.g. will this cause more floods? The rather abstract mm values could be interpreted within exceedance probabilities or so to make sense of them. This should not only be discussed as a by-product but analysed and demonstrated. Such a focus would require a thorough analysis and discussion of how the downscaling and bias-correction affect the results - are they able to reproduce and project daily to weekly joint warm and moist events in winter such as for example the atmospheric rivers that are mentioned? I am a little skeptic how an analogues procedure will still be concurrent with the climate model projection trends at daily scale then. But this could be analysed.

Another option may indeed be to focus on key features of river flow variability that are important for salmon. In any case, a clear focus and message will be required that will make readers remember more than 'again a general shift from snow to more rain-dominated regime in winter'. The necessary revisions may be too substantial to be considered the same paper, but it could perhpas be resubmitted with a more focused title and content to the same Special Issue.

[Figure]

Overall nice figures. Small comments: Figure 7 - good start of this and illustrative, but is the absolute amount of the variability (scale) really so relevant? For readers who don't know the river... Figure 6 - right panels should perhaps use another color scheme. I found the same to be confusing.

---

## Author Comment (AC3) · 25 Jul 2018

**Response to Reviewer # 2 specific comments**

a. p.5 l.3-4: While the authors state that it "[...] is important to evaluate such regime transitions on regional scales while characterizing snowmelt and rainfall driven flows independently", they never clearly state why this is important and how they will use this analysis.

*a. The FRB exhibits substantial spatial variation of air temperature and precipitation due to its complex topography and maritime influences. The hydrologic response therefore varies considerably across the basin differentiating its flows mainly into snow-dominant or hybrid (rain and snow) regimes. These distinct flow regimes are expected to change under future climate change with hybrid or rainfall-dominant flow regimes becoming more prevalent. Such changes will most probably accelerate earlier onset of spring snowmelt and will increase magnitude of summer flood events in snowmelt-dominant flow regimes and will increase winter flows and flood events in rainfall-dominant flow regimes. Therefore the quantification of flow regime transitioning is extremely important for the FRB that could have implications for reginal adaptation measures and water resources management in the region.*

*To quantify such changes, we have used the snowmelt pulse detection technique to characterize snowmelt and rainfall driven flows independently. This technique clearly filters snowmelt-dominant flows from rainfall-dominant flows using the maximum cumulative departure within the define time window (Fig. 3 in the manuscript). In our revised manuscript, we have included a paragraph describing importance of flow regime transitioning in section 4.3. We have also further clarified snowmelt pulse detection technique in section 2.5 of the revised manuscript.*

b. Section 2: This section should reference their earlier work (Islam et al., 2017) more directly, as much of the model setup is the same, in particular the setup of the hydrological model. As is, the section is rather uneven. It goes into great detail regarding the resolution of the ANUSPLIN dataset ("having a spatial resolution of about 9.26 km in the meridional direction and one that varies proportionally to the cosine of latitude in the zonal direction.") but says nothing about the VIC calibration or setup. Incidentally - is the ANUSPLIN dataset simply a 5 arcmin resolution (1/12_)?

*b. We have revised section 2 to make it more informative about the VIC model setup and calibration and removing unnecessary details about the ANUSPLIN data. Yes, the ANUSPLIN data are of 5 arcmin resolution.*

c. Section 2.2: In addition to the strengths, the authors should also address the shortcomings of the downscaling techniques that they use, especially since they look at variability in daily time series. For example, Gutmann et al. (2014) noted that BCCA overestimates wet day fraction and underestimates extreme events. Perhaps the combination with BCCI fixes this, but that would be good to discuss.

*c. BCCAQ2 does, in fact, reduce the magnitude biases in BCCA, as pointed out in the Werner & Cannon (2016) reference given in our Sec. 2.2 (p. 8). More specifically, as reported in a recent paper by Li et al. (2018), "BCCAQ avoids these issues by separating the downscaling and bias correction operations: BCCA, which includes a quantile mapping step at the GCM scale and subsequently generates realistic fine-scale spatial variability, precedes the*

*application of second quantile mapping at each grid point to further correct quantile distributions at the fine scale. Furthermore, the quantile mapping algorithm that is used explicitly preserves the climate change signal additively for temperature and multiplicatively for precipitation of the underlying climate model projections (Cannon et al. 2015)." According to Werner & Cannon (2016), BCCAQ "really shone for use with modelling hydrologic extremes. In this context, it exceeded all other methods." Further, as modified via BCCAQ2, it is especially suited to climate change applications, as featured in Cannon et al. (2015). That said, any bias correction method is only as good as the target data set used, and in this respect, the known biases of ANUSPLIN (e.g., the low precipitation bias at high elevations) are of course transmitted to the downscaled model results via BCCAQ2. This is a point we have now made explicitly in Sec. 2.2 of the revised manuscript, and added additional references as necessary.*

d. p.9 l.1: Wu et al. (2011) does not describe a routing scheme, but simply provides routing networks at different spatial resolutions. From the sentence that follows it appears that the authors have used the Lohmann routing scheme. This should be clarified.

*d. Thanks for pointing this out. We have now revised these sentences and have clearly stated that the Lohmann et al. routing scheme was used to extract runoff at basin outlet.*

e. Section 2.3: The authors do not provide sufficient detail about the VIC setup. It is fine to refer to their earlier paper, but it would be good to mention model resolution, a two-line summary of the source of the parameters, etc. That would be more useful than the long list of references to previous uses of the VIC model (p. 9 second paragraph).

*e. In our revised manuscript, we have now included a paragraph under section 2.3 describing the VIC model resolution, parameters and it calibration and validation for the FRB.*

f. p.11 l.1-2: "Peak runoff during the cold season was computed between 1 October and 1 March when the 3-day running mean daily air temperature exceeds 0 C at each gridcell." Why the extra condition based on air temperature?

*f. Using the extra condition based on air temperature helps to identify the end of the cold season more precisely in each year. The last day of the cold season therefore depends on the temperature criterion. A 3-day running mean is used to avoid extreme events when daily mean temperature exceeds 0°C for a given day within the cold season. In our revised manuscript, we have clarified this point under section 2.4.1.*

g. Section 2.4.2: This section needs to be streamlined. The equations are unnecessary, since most of us know how to calculate a mean and variance for a data set.

*g. In our revised manuscript, we have removed these equations and have explicitly defined all the symbols.*

h. In the results section I found the narrative hard to follow in part because of the way in which the authors use abbreviations to refer to the different sub-basins. Sentences such as "The advance in the timing of the annual peak flow in these sub-basins is slightly less than for the FRB as a whole (_20 days for UF, _18 days for QU, _25 days for TN and _35 days for CH) [...]" are

difficult to read. The numbers may be more effectively presented in a Table, which allows the text to focus on some particular insight that can be derived from this.

***h. As per the Reviewer's suggestion, we have deleted this parenthesized portion in the revised manuscript, and have included these numbers in the form of a table in the supplementary document.***

i. Figures were generally of good quality.

***i. Thank you.***

**References:**

*Cannon, A. J., Sobie, S. R. and Murdock, T. Q.: Bias correction of GCM precipitation by quantile mapping: How well do methods preserve changes in quantiles and extremes? J. Climate, 28, 6938-6959, 2015.*

*Li, G., Zhang, X., Cannon, A. J., Murdock, T., Sobie, S., Zwiers, F., Anderson, K. and Qian, B.: Indices of Canada's future climate for general and agricultural adaptation applications, Clim. Change, 148(1–2), 249–263, doi:10.1007/s10584-018-2199-x, 2018.*

*Werner, A. T. and Cannon, A. J.: Hydrologic extremes - an intercomparison of multiple gridded statistical downscaling methods. Hydrol. Earth Syst. Sci., 20, 1483–1508, 2016.*

---

## Author Response (AR1)

**Responses to the Editor and Reviewers**

*We thank the Reviewers and the Editor for their constructive comments, which helped us to improve the manuscript. We provide our detailed responses (in bold italic) to all the comments submitted by the editor and each referee along with the information on how the paper is revised as per the anonymous referees' suggestions.*

**Editor:**

Based on the referee comments and suggestions, the manuscript needs major revision to bring it up to the standards required for publication, but I do believe the paper has the potential to make a new and important contribution. Your responses indicate a willingness to accept most of their advice. There were a few overarching concerns and issues that should be addressed. First is that the paper needs to clearly demonstrate what is novel and unique. There should be emphasis on how this relates to other regions, what insights are transferable, and how this advances our understanding and prediction of hydrological change in cold, mountainous regions under a warming climate. It should be made clear how this study differs and builds upon the earlier work here by the lead author (stated in the introduction), and also where this work is leading towards (i.e. the second paper to look at extremes).

*Thank you for handling our manuscript and allowing us to revise it as per the reviewers' suggestions. The revised manuscript was restructured considerably to address the reviewers' comments and overarching concerns you mentioned above. Most of the sections in the manuscript are re-written to better highlight the novel aspects of our work. The revised discussion and conclusion sections now provide information on the transferability and importance of our findings to other mid-latitude, mountainous basins with strong maritime influences. In our revised Results section, we have provided a more comprehensive analysis of future-projected hydrological changes to i) estimate specific processes that contribute most to the increase in runoff variability and to ii) explain why the signal of increased future runoff is so much more evident in the Coast Mountains region. The later utilizes a multivariate linear regression analysis to identify the key driver(s) that control changes in runoff mean under projected climate change. Furthermore, in the revision process, we have emphasized how the methodology and the research focus of the present work differ from the earlier study of Islam et al. (2017). Specifically, we have highlighted the methodological improvements of the present study over Islam et al. (2017) by including further details of modifications applied to the new downscaling and bias correction scheme and the utility of the snowmelt detection algorithm.*

*Regarding your point "where this work is leading towards", we have now mentioned explicitly in the introduction section that the present paper is the first of two papers analyzing the same set of hydroclimatic simulations. The present effort deals with features of the transition from seasonal snow to a hybrid snowmelt/rainfall runoff regime, with special attention to the changes in snowmelt dynamics and daily runoff variability. A forthcoming paper (in preparation) addresses the consequences of these changes for river discharge at the main outlet to the FRB at Hope, BC, including a formal flood frequency (extreme value) analysis for the 21st century.*

A concern is the fact that land cover change is not explicitly dealt with or addressed, as noted by reviewer #1. Indeed, when applying models of today under future climates, "turning handles", and getting results, it is questionable how meaningful or useful the results really are. It may be premature to say that land cover change and deglaciation constitute a second order forcing – these could potentially have a major influence and unanticipated consequences. In this region, under all plausible future climate scenarios, we expect rapid deglaciation in the headwaters and widespread forest impacts across the basin due to changing fire regime and other disturbances. How this will affect runoff responses and land–atmosphere interactions is not fully understood. Although it may be beyond the scope of this study to include scenarios of land cover change, this is a major limitation of the study and needs to be properly considered and addressed in the discussion.

*We agree that the land cover changes are not explicitly addressed in our initial manuscript. In the revised discussion section, we have included a detailed discussion focusing on the potential impact of land use and glaciers on hydrological simulations along with clarification that the present effort deals solely with projected hydrological changes under strong greenhouse gas forcing. Once the primary hydrologic response to this forcing has been estimated, a follow-up study focused on the effects of land cover change and glacier retreat (with a hydrologic model including dynamic glaciers) will be warranted.*

**Reviewer # 1**

In this study, the authors examined the influence of future climate scenarios on streamflow in the Fraser River basin in British Columbia, Canada. They used statistically downscaled output from 21 GCMs for the RCP 8.5 emissions scenario, using one realization from each GCM. The authors used the VIC hydrologic model, which has been applied in previous studies to look at the effects of climate and land-cover change on streamflow. Key results are that the basin will transition from a snow-dominated regime to a more rain-dominated regime, and that flow variability will increase in winter, with an increase in the magnitude of cold-season peak flows.

*Thank you kindly for reviewing this manuscript.*

**1.** Overall, the study appears to have been conducted in a competent manner using up to-date approaches for generating the future climate scenarios. I expect that the results will be of great interest to the agencies involved in managing water-related resources and hazards in the Fraser basin. However, the manuscript reads like a regional case study, and I struggled to discern how this work contributes novel and significant knowledge in the context of the international readership of HESS.

*1. We appreciate the careful review of our manuscript, and thank the Reviewer for his/her perceptive comments. While a majority of HESS papers are in fact regionally focused, we agree that we have not paid sufficient attention to what this particular "regional case study" can teach us about other similar regions in the world. In fact, as we discuss below, the somewhat unusual physical setting of the FRB can teach us a great deal about hydroclimatic change in a mid-latitude, mountainous basin with strong maritime influences. Furthermore, as our manuscript is targeted for the HESS special issue on "Understanding and prediction earth system and hydrological changes in cold regions," in our revised manuscript we have distinguished projected changes in the FRB that are fairly universal from those that are case-specific. We have revised the discussion section in our revised manuscript to highlight this point.*

2. The shift from snow-to rain-dominated regimes in mountainous mid-latitude catchments has been identified in dozens, if not hundreds, of earlier climate-impact studies published in the international literature.

*2. We agree that there are many studies reporting snow- to rain-dominated regime changes. However, the FRB does in fact differ in important ways from other mountainous, mid-latitude catchments. While it is indeed a mid-latitude, nival basin, it extends from the Pacific coast to the continental interior, meaning that it is also maritime influenced. Specifically, the hydrologic response to warming in the FRB is influenced by two possibly confounding factors: first, the change of phase from snow to rain; and second, the very significant increase in atmospheric moisture supply (atmospheric rivers) to the North American west coast as anticipated in the CMIP5 future projections (Payne & Magnusdottir, 2015; Radic et al., 2015; Warner et al., 2015; Warner & Mass, 2017). While small, mountainous catchments on the Norwegian coast are also strongly influenced by atmospheric rivers, they have hybrid pluvial-nival regimes at present, and lack the extensive interior snowpack that exists in the FRB headwaters of the Canadian Rockies. An important research question that we aim to address is*

*whether this interior snowpack will increase or decrease in response to these large-scale changes, because the exact, geographic and seasonal change in moisture supply needs to be included as part of the modelling chain. In the revised manuscript and Supplemental Material, we have now included results (Figure 3, Table 2, and Supplementary Figure 5) that make the "added value" of our simulation strategy for answering these types of questions much more evident.*

3. Based on descriptions of the model set-up in earlier work by the authors, I infer that land cover was held constant through the simulations. In reality, however, land-cover will evolve, particularly in response to widespread forest disturbance related to the Mountain Pine Beetle outbreak that began in the 1990s, and the salvage logging that followed. In addition, glacier retreat will undoubtedly influence the hydrology of some of the mountainous headwaters. An important question is the extent to which these land-cover changes would amplify or diminish the effects of climatic change.

*3. While an investigation of land cover effects might be of interest, it is beyond the scope of the present effort, which deals solely with the impacts of projected changes in climate on the FRB's cold season runoff variability and flow regimes under strong greenhouse gas forcing. We point out that any choice of land use / land cover scenario is arbitrary considering that all future changes are conditional on a given scenario. In addition, it would be difficult to predict how forest composition and disturbance regimes will change in the future. We also suspect that the effect of glacier losses and ultimately the end of glacier wasting will not have a fundamental impact on future flow regimes at the scale of the FRB. In fact the hydrological model used in our study does store and ablate frozen water at high elevations in the form of piles of snow that grow over time under historical climate conditions. While those snow piles do not have quite the same albedo and other surface properties as ice and they do not flow, they represent crude glaciers in model simulations that grow during the historical period in some locations, and subsequently ablate as melting outpaces deposition.*

*Most importantly, it is likely that both land cover change and glacier retreat constitute second-order forcings compared to the dominant effect of the strong increase in greenhouse gas forcing under the RCP8.5 scenario. Once the primary hydrologic response to RCP8.5 has been estimated, a follow-up study focused on the effects of land cover change and glacier retreat (with a hydrologic model including dynamic glaciers) will be warranted.*

*In our revised manuscript, we have included further detail in the discussion section to address these issues.*

4. On balance, I am not fundamentally opposed to the publication of this work, but I believe the authors need to make a more convincing case that this study represents an internationally significant contribution to the literature and is not just a regional case study. The authors need to highlight what is novel about this work when considered within the broader context of the international literature. I should note that I have not kept up with the climate-impacts literature for a few years, and I may not have the background to appreciate the novelty of this work without it being spelled out more explicitly.

*4. We thank the Reviewer for this suggestion and reiterate that our revision has better highlighted the novel aspects of this work. In our revised Results and Discussion sections, we have provided a more comprehensive analysis of future-projected hydrological changes in the FRB (described in detail in our response to point 1 of Reviewer 2) that clearly sets our work apart from that on other mountainous, mid-latitude catchments in the published literature.*

**Reviewer # 2**

This is a generally well-written paper that discusses changes in runoff variability and flow regimes in the Frasier River Basin under climate change. The authors analyze 21 downscaled CMIP5 simulations that have been used as input to a VIC model implementation at 0.25 resolution. While the paper is generally well-written, the study itself is mostly routine and is not sufficiently novel in its current form that I can recommend publication in HESS.

*We thank the Reviewer for providing insightful comments and suggestions on our manuscript. The key issue touched on here regarding the novelty of our work is addressed in detail under point 2 below.*

1. The paper is purely descriptive in its analysis. The authors describe the results from the model simulations, but make no real attempt to analyze and interpret them. For example, which specific processes contribute most to the increase in runoff variability? How does this increased variability affect the salmon population (the authors repeatedly make the point that the Frasier supports the largest migration of Pacific Ocean sockeye salmon in the world)?

*1. Thank you for raising this issue. We agree with the referee's concerns and have placed additional emphasis on the analysis and explanation of model results throughout the Results and Discussion sections. In addition, we performed additional analyses to address the Reviewer's question, "which specific processes contribute most to the increase in runoff variability?" First, we examined the rate of change of runoff variability and mean with respect to the amount of warming in the Coast Mountains, Interior Plateau and Rocky Mountains subregions of the FRB (Figure 3). In the Coast Mountains, we found that the change in cold season mean runoff is significantly larger than the change in runoff variability, unlike in the other two regions. This finding helps to explain why the signal of increased future runoff is so much more evident in the Coast Mountains region. Second, we employed a multivariate linear regression model to decompose the cold season runoff monthly variability into separate contributions from precipitation phases (rainfall and snowfall) and temperature (Table 2 and Supplementary Figure 5). The model, which explains 50-90% of the total variance in runoff over a large portion of the basin, allows us to estimate the contributions of these drivers to the simulated runoff variability at individual gridcells.*

*The question of impacts on the salmon population is beyond the scope of this study considering that such an impact assessment would involve an examination of water temperature, oxygen content, and also a solid knowledge of salmonid biology, which is not our area of expertise. We have, however, initiated a study to estimate the impacts of changing water temperatures on salmon migration and populations in the Fraser River, which will be reported in a future publication.*

2. As is, the findings mostly have regional interest, as there is no new methodological development nor are any of the findings particularly surprising. Warming in climates with a large seasonal snow component will lead to larger flows in winter because of a shift from snowfall to rain, mid-season melt, and earlier melt, but this has been widely reported for similar basins in western North America and Europe. The field has progressed to where this may be extremely

useful information for local water managers, but the study design and findings by themselves are not sufficient for publication in a scientific journal.

*2. We thank the Reviewer for raising these points, which are similar to those raised by Reviewer #1. With regard to the comment, "the findings mostly have regional interest," we point out that, in fact, most papers published in HESS have a regional focus. However, we agree that we did not pay sufficient attention to what our study of the FRB could teach us about other similar regions around the world. As mentioned in the response to Reviewer #1's point 2, we agree that we could better highlight the novel aspects of our work, and have made several changes to the revised manuscript to do just that. Previous studies that examined future hydroclimatic changes in the FRB were mostly focused on monthly and annual time scale differences in mean climatology and hydrographs. By contrast, there has been relatively little work quantifying cold season, daily time scale flow variability and regime transitions in the FRB. These research goals necessitate the use of a large model ensemble, an effective downscaling and bias-correction method, and a robust snowmelt detection algorithm. Our determination of snowmelt-dominant categories (SDCs) and their future change, carried out at fine spatial scales in Section 3.3, was not mentioned by either Reviewer. Yet, this is to our knowledge an original contribution in a hydroclimatic modelling context for any basin worldwide to study projected runoff regime transition. This study is therefore not a routine effort and represents a significant advance over what has appeared in the published literature. Nevertheless, we are strongly motivated by the Reviewer's comment to better emphasize the novel methodology and key research results of this study. We have revised the methodology section accordingly.*

*Regarding the Reviewer's comment that, "Warming in climates with a large seasonal snow component will lead to larger flows in winter because of a shift from snowfall to rain, mid-season melt, and earlier melt, …", we agree that most prior research bears this out. However, the situation is not so simple in the FRB, since there is evidence of a significant increase in atmospheric rivers impacting the North American west coast as projected in the CMIP5 models. In the revised manuscript and Supplemental Material, we have included new results that make the added value of our simulation strategy for answering these types of questions much more evident.*

3. The paper is not significantly different from an earlier paper by the same lead author in Journal of Hydrometeorology (doi:10.1175/JHM-D-16-0012.1), which discusses the same modeling chain and setup (some of the figures are near identical and should be attributed at the very least). That paper is based on a smaller model ensemble and focuses more on changes in the mean climate / hydrology rather than changes in variability. If the authors choose to focus on variability in this paper, then I would encourage them to analyze what this increase in variability actually means for the basin.

*3. In fact, as mentioned above, both the methodology and the research focus of the present work differ significantly from the earlier study of Islam et al. (2017), although we acknowledge that these aspects were not sufficiently emphasized in the submitted manuscript. In the Methods section of the revised paper, we have highlighted the methodological improvements of BCCAQ2 downscaling and bias correction over that of BCSD, along with the utility of the snowmelt detection algorithm we employed. Specifically, we emphasize that*

*future-projected changes in daily flow variability and runoff extremes cannot be accurately examined using BCSD-downscaled driving data. We also point out that Islam et al. (2017) examined projections only out to the 2050s using 12 CMIP5 models, while we use a 21-model CMIP5 ensemble and provide projections to 2100. The use of 21 models in this study allows us to better sample the uncertainty in driving GCMs. Indeed, the use of BCCAQ2 and the extended time horizon lead to new insights into projected changes in regional runoff and its variability that are not previously available in literature (e.g., the strongly increasing peak runoff in the Coast Mountains after ~2040).*

4. The manuscript lacks a clear conclusion section as the authors have a single combined discussion and conclusion without a clear take-home message. I would suggest splitting these components as it emphasizes the need to have a clear conclusion that adds to the existing body of knowledge and that is focused on findings that are of wider interest than the local changes in the Frasier River basin.

*4. In our submitted manuscript, a separate Conclusions section has been created after the Discussion, to provide readers with a succinct and clear take home message.*

Specific comments:

a. p.5 l.3-4: While the authors state that it "[...] is important to evaluate such regime transitions on regional scales while characterizing snowmelt and rainfall driven flows independently", they never clearly state why this is important and how they will use this analysis.

*a. The FRB exhibits substantial spatial variation of air temperature and precipitation due to its complex topography and maritime influences. The hydrologic response therefore varies considerably across the basin differentiating its flows mainly into snow-dominant or hybrid (rain and snow) regimes. These distinct flow regimes are expected to change under future climate change with hybrid or rainfall-dominant flow regimes becoming more prevalent. Such changes will most probably accelerate the onset of spring snowmelt and will modulate the magnitude of summer flood events in snowmelt-dominant flow regimes and will increase winter flows and flood events in rainfall-dominant flow regimes. Therefore the quantification of flow regime transitioning is particularly important for the FRB that could have implications for reginal adaptation measures and water resources management in the region.*

*To quantify such changes, we have used the snowmelt pulse detection technique to characterize snowmelt and rainfall driven flows independently. This technique separates snowmelt-dominant flows from rainfall-dominant flows using the maximum cumulative departure within the defined time window (Supplementary Fig. 1). In our revised manuscript, we have included a further discussion of flow regime transitioning in the discussion and conclusion sections. We have also further clarified the snowmelt pulse detection technique in Section 2.5 of the revised manuscript.*

b. Section 2: This section should reference their earlier work (Islam et al., 2017) more directly, as much of the model setup is the same, in particular the setup of the hydrological model. As is, the section is rather uneven. It goes into great detail regarding the resolution of the ANUSPLIN dataset ("having a spatial resolution of about 9.26 km in the meridional direction and one that

varies proportionally to the cosine of latitude in the zonal direction.") but says nothing about the VIC calibration or setup. Incidentally - is the ANUSPLIN dataset simply a 5 arcmin resolution (1/12_)?

*b. We have revised Section 2 to make it more informative about the VIC model setup and calibration and removing unnecessary details about the ANUSPLIN data. Yes, the ANUSPLIN data are of 5 arcmin resolution.*

c. Section 2.2: In addition to the strengths, the authors should also address the shortcomings of the downscaling techniques that they use, especially since they look at variability in daily time series. For example, Gutmann et al. (2014) noted that BCCA overestimates wet day fraction and underestimates extreme events. Perhaps the combination with BCCI fixes this, but that would be good to discuss.

*c. BCCAQ2 does, in fact, reduce the magnitude of biases in BCCA, as pointed out in the Werner & Cannon (2016) reference given in our Sec. 2.2 (p. 8). More specifically, as reported in a recent paper by Li et al. (2018), "BCCAQ avoids these issues by separating the downscaling and bias correction operations: BCCA, which includes a quantile mapping step at the GCM scale and subsequently generates realistic fine-scale spatial variability, precedes the application of second quantile mapping at each grid point to further correct quantile distributions at the fine scale. Furthermore, the quantile mapping algorithm that is used explicitly preserves the climate change signal additively for temperature and multiplicatively for precipitation of the underlying climate model projections (Cannon et al. 2015)." According to Werner & Cannon (2016), BCCAQ "really shone for use with modelling hydrologic extremes. In this context, it exceeded all other methods." Further, as modified via BCCAQ2, it is especially suited to climate change applications, as featured in Cannon et al. (2015). That said, any bias correction method is only as good as the target data set used, and in this respect, the known biases of ANUSPLIN (e.g., the low precipitation bias at high elevations) are of course transmitted to the downscaled model results via BCCAQ2. This is a point we have now made explicitly in Sec. 2.2 of the revised manuscript, and added additional references as necessary.*

d. p.9 l.1: Wu et al. (2011) does not describe a routing scheme, but simply provides routing networks at different spatial resolutions. From the sentence that follows it appears that the authors have used the Lohmann routing scheme. This should be clarified.

*d. Thanks for pointing this out. We have now revised these sentences and have clearly stated that the Lohmann et al. (1996, 1998a, b) routing scheme was used to extract runoff at basin outlets.*

e. Section 2.3: The authors do not provide sufficient detail about the VIC setup. It is fine to refer to their earlier paper, but it would be good to mention model resolution, a two-line summary of the source of the parameters, etc. That would be more useful than the long list of references to previous uses of the VIC model (p. 9 second paragraph).

*e. In our revised manuscript, we have now included paragraphs under section 2.3 describing the VIC model resolution, parameters, and it calibration and validation for the FRB.*

f. p.11 l.1-2: "Peak runoff during the cold season was computed between 1 October and 1 March when the 3-day running mean daily air temperature exceeds 0 C at each gridcell." Why the extra condition based on air temperature?

*f. Using the extra condition based on air temperature helps to identify the end of the cold season more precisely in each year. The last day of the cold season therefore depends on the temperature criterion. A 3-day running mean is used to avoid extreme events when daily mean temperature exceeds $0^\bullet$C for a given day within the cold season. In our revised manuscript, we have clarified this point under section 2.4.1.*

g. Section 2.4.2: This section needs to be streamlined. The equations are unnecessary, since most of us know how to calculate a mean and variance for a data set.

*g. In our revised manuscript, we have removed these equations and have explicitly defined all the symbols.*

h. In the results section I found the narrative hard to follow in part because of the way in which the authors use abbreviations to refer to the different sub-basins. Sentences such as "The advance in the timing of the annual peak flow in these sub-basins is slightly less than for the FRB as a whole (_20 days for UF, _18 days for QU, _25 days for TN and _35 days for CH) [...]" are difficult to read. The numbers may be more effectively presented in a Table, which allows the text to focus on some particular insight that can be derived from this.

*h. As per the Reviewer's suggestion, we have deleted this parenthesized portion in the revised manuscript, and have included these numbers in Table 1.*

i. Figures were generally of good quality.

*i. Thank you.*

**Reviewer # 3**

*We thank the Reviewer for the careful review of our manuscript and useful suggestions to further improve the analysis.*

1. My first assessment was similar to that of the previous reviewers: what has been modelled for the Fraser River has been modelled and reported many times before: changes in mean flow, regime, snow-rain ratio, etc. Abstract and conclusion provide little new information and the international reader doesn't know what knowledge gain to transfer to other regions. In this context we should remember that HESS has the same requirements for special issue papers as for regular contributions. Manuscripts submitted as type 'research articles' should 'clearly advance our understanding', ms type 'cutting edge-case study' needs to provide all data to serve others as testbed e.g. for models (from the HESS website). The current manuscript is perhaps in-between.

*1. We agree that there are many studies reporting changes in mean flow and snow- to rain-dominated regime changes in the FRB. However, most of these studies have examined future hydroclimatic changes on monthly and annual time scales focusing on the spring and summer seasons. By contrast, there has been relatively little work quantifying cold season, daily time scale flow variability and regime transitions in the FRB. These research goals require the use of a large model ensemble, an effective downscaling and bias-correction method (BCCAQ2), and a robust snowmelt detection algorithm. Furthermore, our determination of snowmelt-dominant categories carried out at fine spatial scales, is an original contribution in a hydroclimatic modelling context for any basin worldwide to study projected runoff regime transitions. This study is therefore not a routine effort and represents a significant advance over what has appeared in the published literature. Nevertheless, we are strongly motivated by all the Reviewers' comments to better emphasize the novel methodology and key research results of this study in the revised manuscript.*

*Regarding the Reviewer's comment on "what knowledge gain to transfer to other regions", we agree that, in the original submission, we did not pay sufficient attention to what this particular regional study can teach us about other similar regions in the world. As mentioned in our response to the other reviewers, the specific physical setting of the FRB can teach us a great deal about hydroclimatic change in a mid-latitude, mountainous basin with strong maritime influences. In our revised manuscript, we have distinguished projected changes in the FRB that are fairly universal from those that are case-specific.*

2. A symptomatic indicator is the start of Section 5 "..overall question...how...precipitation phase and variability will modulate the FRB's runoff variability and flow regimes". Instead of this case study view, the science question should be how cold climate hydrology transitions to temperate climate hydrology - the FRM just happens to be considered the case that is used for illustration. However, with the running model at hand and gauging from the responses given already there is potential to focus on a particular process or phenomenon that is not yet well understood and is still specific to cold regions transitioning to temperate climate.

*2. Thank you for providing this comment on the scientific focus of our study. While our geographic region of focus is the FRB, we have made clear in the revised manuscript that the overall context of the work is indeed just the type of hydrologic transition that the Reviewer*

*identifies. We agree with the Reviewer that the application of the CMIP5-VIC ensemble over the FRB specifically permits us to make a detailed analysis of the processes responsible for this transition.*

3. Some of the analyses on the variability and pulses etc. that are presented here stand out and may provide a nice starting point. They are the ones that could be made the sole focus, analysed more specifically and quantitatively to make this an original contribution specifically dealing with features of the transition from seasonal snow to more rainfall-runoff dominated flow dynamics. It would have been very interesting, for example, to see the analysis on the daily to weekly variability expanded more systematically to scale and quantities - e.g. will this cause more floods? The rather abstract mm values could be interpreted within exceedance probabilities or so to make sense of them. This should not only be discussed as a by-product but analysed and demonstrated. Such a focus would require a thorough analysis and discussion of how the downscaling and bias-correction affect the results - are they able to reproduce and project daily to weekly joint warm and moist events in winter such as for example the atmospheric rivers that are mentioned? I am a little skeptic how an analogues procedure will still be concurrent with the climate model projection trends at daily scale then. But this could be analysed.

*3. The Reviewer should be made aware that the present work is the first of two papers analyzing the same set of CMIP5-VIC simulations. This point is now mentioned explicitly in the introduction section of the revised manuscript. The present paper deals with features of the transition from seasonal snow to a hybrid snowmelt/rainfall runoff regime, with special attention to the changes in snowmelt dynamics and daily runoff variability. Our upcoming paper (Curry et al., in preparation) addresses the point raised by the Reviewer, namely the consequences of these changes for river discharge at the main outlet to the FRB at Hope, BC, including a formal flood frequency (extreme value) analysis for the 21st century. The methodological question raised by the Reviewer "how the downscaling and bias-correction affect the extreme value analysis" is also addressed in this forthcoming work.*

*As discussed in our response to the other reviewers, in the revised manuscript we have conducted additional analysis to address the issue raised by the Reviewer, namely "features of the transition from seasonal snow to more rainfall-runoff dominated flow dynamics". The new analysis has allowed us to determine the rate of change of runoff variability and mean with respect to the amount of warming in the Coast Mountains, Interior Plateau and Rocky Mountains subregions of the FRB (Figure 3). We also employed a multivariate linear regression model to decompose the cold season runoff monthly variability into separate contributions from rainfall, snowfall and air temperature (Table 2 and Supplementary Figure 5). This procedure allows us to estimate the contribution of these drivers to the simulated runoff variability at individual gridcells. These additional analyses, combined with existing results in the paper addressing changes in snow dominant categories and daily runoff variability, both heighten the impact of the present work and provide a foundation for our forthcoming work on streamflow extremes in the FRB.*

4. Another option may indeed be to focus on key features of river flow variability that are important for salmon.

*4. As discussed in the Response to Reviewer #2, point 1, the question of impacts on the salmon population is beyond the scope of this study.*

5. In any case, a clear focus and message will be required that will make readers remember more than 'again a general shift from snow to more rain dominated regime in winter'. The necessary revisions may be too substantial to be considered the same paper, but it could perhaps be resubmitted with a more focused title and content to the same Special Issue.

*5. As mentioned in the response to Reviewers 1 & 2, the Conclusion of our revised manuscript now provides a clear take-home message that better highlights the novel aspects of our work. The additional analysis we have conducted not only provides readers with a clearer notion of the mechanisms behind hydrologic change in the FRB, but also points out commonalities with other mid-latitude basins that are likely susceptible to the same climatic drivers.*

**Minor comments:**

6. Figure 7 - good start of this and illustrative, but is the absolute amount of the variability (scale) really so relevant? For readers who don't know the river.

*The units for all three panels in Figure 7 (Figure 8 in the revised manuscript) are kept identical to maintain consistent and comparable results. The values in panel b and c are daily and 7-day standard deviations. Use of the same units allows readers to clearly see changes in variability, which would be obscured using the Coefficient of Variation (CV) with a changing mean.*

7. Figure 6 - right panels should perhaps use another color scheme. I found the same to be confusing.

*In our revised manuscript, we have changed the color scheme for all three right panels in Figure 6 (Figure 5 in the revised manuscript).*

*Correspondence to*: Siraj Ul Islam (sirajul.islam@unbc.ca)

**Abstract.** In response to ongoing and future-projected global warming, mid-latitude, nival river basins are expected to transition from a snowmelt-dominated flow regime to a nival-pluvial regime with an earlier spring freshet of reduced magnitude. There is, however, a rich variation in responses that depends on factors such as the topographic complexity of the basin and the strength of maritime influences. We illustrate the potential effects of a strong maritime influence by studying future changes in cold season flow variability in the Fraser River Basin (FRB) of British Columbia, a large extratropical watershed extending from the Rocky Mountains to the Pacific Coast. We use a process-based hydrological model driven by an ensemble of 21 statistically downscaled simulations from the Coupled Model Intercomparison Project Phase 5 (CMIP5) following the Representative Concentration Pathway (RCP) 8.5.

Warming under RCP8.5 leads to reduced winter snowfall, shortening the average snow accumulation season by about one-third. Despite this, large increases in cold season rainfall lead to unprecedented cold season peak flows and increased overall runoff variability in the VIC simulations. Multivariate linear regression analysis further reveals a rainfall increase in the cold season as a

dominant climatic driver in the Coast Mountains contributing 60% to mean runoff change in the 2080s. At the main outlet to the basin, cold season runoff increases by 70% by the 2080s and its interannual variability more than doubles compared to the 1990s, suggesting substantial challenges for operational flow forecasting in the region. Furthermore, the application of a snowmelt pulse detection algorithm classifies the entire FRB as a snow-dominated runoff regime in the 1990s with 45% of the basin area transitioning to primarily rain-dominated in the 2080s. While these projections are consistent with the anticipated transition from nival to nival/pluvial in the FRB, the marked increase in cold season runoff is likely linked to more frequent landfalling atmospheric rivers in the region projected in the CMIP5 models, providing insights for other maritime-influenced extratropical basins. Canada's Fraser River Basin (FRB), the largest watershed in the province of British Columbia, supplies vital freshwater resources and is the world's most productive salmon river system. We evaluate projected changes in the FRB's runoff variability and regime transitions using the Variable Infiltration Capacity (VIC) hydrological model. The VIC model is driven by an ensemble of 21 statistically downscaled simulations from the Coupled Model Intercomparison Project Phase 5 (CMIP5), for a 150-year time period (1950-2099) over which greenhouse gas concentrations follow the CMIP5 Representative Concentration Pathway (RCP) 8.5. Using mean and standard deviation (variability) metrics, we emphasize projected hydroclimatological changes in the cold season (October to March) over different sub-basins and geoclimatic regions of the FRB.

Warming consistent with the RCP8.5 scenario would lead to increased precipitation input to the basin with higher interannual variability and considerably reduced winter snowfall shortening the average snow accumulation season by about 38%. Such changes in temperature and precipitation will

increase cold season runoff variability leading to higher cold season peak flows. In the lower Fraser River, cold season runoff will increase by 70% and its interannual variability will double compared to the 1990s, presenting substantial challenges for operational flow forecasting by the end of this century. Cold season peak flows will increase substantially, particularly in the Coast Mountains, where the peak flow magnitudes will rise by 60%. These projected changes are consistent with a basin-wide transition from a snow-melt driven flow regime to one that more closely resembles a rainfall driven regime. This study provides key information relating to projected hydroclimate variability across the FRB, describes potential impacts on its water resources, and assesses the implications for future extreme hydrological events.

**Keywords:** Climate change, hydrological modelling, runoff, snow,  mid-latitude basin, Fraser River Basin

**1 Introduction**

Climate change will continue to modify the hydrology of mid-latitude, mountainous river basins especially those with strong maritime influences. The hydrologic response to increasing temperature and changing precipitation phase will be quite distinct in these basins depending mostly on their topographic features and intensifying maritime influences. One such basin is the  Fraser River Basin (FRB) in the province of British Columbia, Canada, one of  the largest watersheds draining the western Cordillera of North America (Benke and Cushing, 2005). Extending from the Pacific coast to the continental interior, it spans 240,000 km$^2$ of diverse landscapes including dry interior plateaus bounded by the Rocky Mountains to the east and the maritime influenced Coast Mountains  in the west. Its elevation ranges from sea level to 3954 m at its tallest peak, Mt. Robson in the Rocky Mountains (Benke and Cushing, 2005). Descending at Fraser Pass near Blackrock Mountain, the Fraser River runs 1400 km before draining into the Strait of Georgia and Salish Sea at Vancouver, British Columbia (BC) (Schnorbus et al., 2010). ~~The FRB's rich diversity and abundance of natural resources yield economic opportunities and prosperity related to agriculture, fishing, forestry, mining, hydropower production, tourism and recreation (Benke and Cushing, 2005). As an extensive freshwater aquatic ecosystem, it remains the most important salmonid producing river system in North America, sustaining the largest migrations of Pacific Ocean sockeye salmon in the world (Labelle, 2009; Eliason et al., 2011). The Fraser River also forms the aquatic habitat of North America's largest wild population of white sturgeon now listed as an endangered species due to its declining populations (Scott and Crossman, 1973; McAdam et al., 2005; Hildebrand 
[revised manuscript text omitted]
 having at a spatial resolution of 0.0833° (~about 109.26 km × 10 km, depending on latitude) in the meridional direction and one that varies proportionally to the cosine of latitude in the zonal direction. The CMIP5 projections were initially downscaled from the native GCM resolutions to the resolution of ANUSPLIN for the bias correction step.Daily wind speed grids are interpolated from coarse scale (2.5° latitude × 2.5° longitude) daily wind speeds at a 10 m height above ground from the National Centers for Environmental Prediction/National Center for Atmospheric Research (NCEP/NCAR) Reanalysis

[revised manuscript text omitted]

$$\overline{x_i} = \frac{1}{T}\sum_{t=1}^{T} x_{it} \qquad\qquad (1)$$

where $x$ is the variable of interest, $i$ indicates ensemble member and $t$ represents time. The interannual standard deviation $S_i$ for each model was calculated as:

$$S_i = \left[\frac{1}{T-1}\sum_{t=1}^{T}(x_{it}-\overline{x_i})^2\right]^{\frac{1}{2}} \qquad\qquad (2)$$

Using Eqs. (1) and (2), the MME mean climatology $\overline{x}$ and MME mean meanss of these quantities, $\overline{\overline{x}}$ and $\overline{S}$ MME standard deviation, $\overline{S}$ wereareas then estimatedthen calculated as:

$$\overline{\overline{x}} = \frac{1}{I}\sum_{i=1}^{I} \overline{x_i} \qquad\qquad (3)$$

$$\overline{S} = \frac{1}{I}\sum_{i=1}^{I} S_i \qquad\qquad (4), \text{ and.}$$

where I = 21 is the number of simulations. tThe inter-model spread in $\overline{x}$ and $\overline{S}$ was, $S_{MME}$ $S_{MME}$ is characterizestimated by thea 5-95% models rangeconfidence interval in $\overline{x_i}$. AEquation (2) was used to calculate annual (or cold season) variability within each 30 year time period was calculated by evaluating the interannual standard deviation for each day of the water year in the 1990s, 2050s and 2080s time periods. Equations (3) and (4) were used for both the MME mean and the MME mean interannual

The effect of transient warming within the 2050s and 2080s periods in the RCP8.5 scenario  is removed by subtracting the least squares linear trend from each time series before calculating its variability. Variability in 7-day runoff  is  computed using a 7-day running mean of the  daily runoff time series.

In an attempt to better understand the contributions of future air temperature and precipitation change to runoff change in the simulations, we use a multivariate linear regression (MLR) analysis, following a similar approach by Kapnick and Delworth (2013). We decompose the cold season runoff $ROF$ into separate contributions from rainfall $Rn$, snowfall $Sn$, mean air temperature $T$ and residual $E$ on a monthly basis as follows:

$$\frac{\Delta ROF_{n,m}}{ROF_{m,1990s}} = a\frac{\Delta Rn}{ROF_{m,1990s}} + b\frac{\Delta Sn}{ROF_{m,1990s}} + c\frac{\Delta T}{ROF_{m,1990s}} + E \qquad (1)$$

where $a$, $b$ and $c$ are regression coefficients corresponding to the rainfall, snowfall and mean air temperature, $m = 1, \dots, 6$ denotes the cold season month (Oct-Mar), and $n = 2070, \dots, 2099$ represents the water year. $ROF_{m,1990s}$ represents the multi-year mean runoff for each month in the 1990s time period. The regression model is fitted for each gridcell and model independently using detrended monthly anomalies from 2070-2099. Spatial averages over the FRB and geoclimatic regions only use gridcells for which the MLR is statistically significant with a p-value $< 0.05$. The relative contribution of each variable to future runoff change is obtained by normalizing the area average of each term in Eq. (1) by the corresponding area-averaged $\Delta ROF$. We only consider the lag-0 correlation between the

driving and response variables considering that monthly time resolution is sufficient to encompass any lags at the local grid scale.

The 5% and 95% lower and upper bounds for 90% confidence intervals of the MME mean were calculated using the standard deviation among ensemble members where the degrees of freedom are the ensemble members (I-1). A t-test statistic (p < 0.05) was used to test the significance of the linear regression slopes.

**2.5 Snowmelt Pulse Detection**

Apart from the analysis of changes in cold season flow variability, we estimate the flow regime transitions that are usually induced by snowpack reduction under increasing summer temperatures. We investigate transitions to new hydrological regimes in the FRB using thWe used the Snowmelt Pulse (SP) detection technique (Cayan et al., 2001; Fritze et al., 2011) to investigate the potential for transitions to new hydrological regimes in the basins of interest. This technique separates snowmelt-dominant flows from rainfall-dominant flows using the maximum cumulative flow departure from mean flow within the defined time window. The SP date, which is defined as the day when the cumulative departure from that water year's mean flow is most negative, provides a way for determining the time at which increased ablation in a snowmelt-dominant basin initiates the transition from low winter base flows to high spring flow (freshet) conditions. While accumulation of flow departures commenced for each water year commences on 1 October, we only considered those SPs occurring between 1 March (water-day 151; (152 for leap years)) and 15 June (water-day 238; (239 for leap years)), the present-day

freshet period, so as to exclude runoff pulses induced by rainfall events. Runoff is rainfall-dominant when the ratio of the area of positive cumulative flow departure (indicating rainfall events) to the area of negative departure between water-days 151 and 238 is greater or equal to unity.

An illustration of the robustness of the algorithm to different river flow regimes using historical data is shown in Supplementary Fig. 1. The application of the algorithm reveals the presence of a SP in the snowmelt-dominant system in all selected years, while for the rainfall-dominant system, SPs are quite rare.

To explore potential regime changes in the FRB, we evaluate and compare the fraction of years for which SPs are recorded within each analysis period (1990s, 2050s, 2080s) at each 0.25° gridcell and for all CMIP5-VIC simulations (with sample size of 21 CMIP-VIC simulations × 30 years = 630 years). In each simulation, each gridcell is classified into one of four snow-dominant categories (SDCs) as defined by Fritze et al. (2011): SDC1 (clearly rain-dominant: SP occurrence in < 30% of water years); SDC2 (mostly rain-dominant, SP occurrence ≥ 30% but < 50%); SDC3 (mostly snowmelt-dominant, SP occurrence ≥ 50% but < 70%); and SDC4 (clearly snowmelt-dominant, SP occurrence ≥ 70%). This allowed spatial comparisons of regime projections for the 2050s and 2080s

and spatial distributions of gridcells in each SDC relative with the regime characteristics of to those from the 1990s base period. Finally, the SDC results awere aggregated by geoclimatic region.

Overall, this study expands on Islam et al. (2017) who used 12 driving GCMs and only considered projections up to the 2050s to quantify changes in the FRB's mean runoff. Here we evaluate projected changes in runoff variability and flow regimes by the end of this century utilizing a set of 21 CMIP5 GCMs downscaled and bias-corrected using an advanced BBCAQ2 method and an efficient snowmelt pulse detection algorithm.

**3 Results**

We examine first examine the projected changes in the mean and interannual variability of precipitation over the different geographic regions of the FRBFRB. Next, we explore the consequences of these changes for runoff means and variability at various temporal and spatial scales and estimate the contribution. of key drivers that control changes in runoff mean. This is followed by a discussion of changing flow regimes over the FRB at regional and sub-basin scales.

**3.1 Projected Changes in Precipitation and Snow-to-Rain Ratio**

The MME mean precipitation, spatially -averaged over the FRB's gridcells, increases rises steadily over the simulation period, increasireachingby nearly 15% in the 2080s relative to the 1990s base period both in annual mean and in the cold season (Fig. 34a, b). The changes are largest in the northern and eastern FRB (Supplementary Fig. 21a, c) reaching up to 20% in the cold season (Supplementary Fig. 21c). The MME mean precipitation interannual variability over the FRB increases by 15%continuously with warming between 2 and 5 °C, then increases more sharply to over 25% as this level of warming is

exceeded (after the 2060s)  (Fig. 3a

). Thus cold season precipitation variability

increases approximately linear  a rate of ~ 4% °C$^{-1}$ towards the end of

the 21$^{st}$ century compared to the 1990s, about double the rate of change of .

 MME mean precipitation. The larger increase in

Precipitation variability

~~~20% to ~80% in the interior and northern FRB (Supplementary Fig. 21d). In the cold season, the~~

compared with mean precipitation is seen

throughout the simulation period for both the entire FRB and its three geoclimatic regions.

Nevertheless, the models' 5-

95%  tend to overlap (except in the Interior Plateau; Fig. 3a, right), reflecting the

considerable spread amongst models.

The partitioning of MME mean total precipitation into rainfall and snowfall reveals substantial

increases in daily rainfall towards the end of the 21$^{st}$ century across the Coast Mountains, Interior

Plateau and Rocky Mountains (Fig. 4a, c, e). The increase in rainfall emerges prominently in the Coast

Mountains in the latter half of the 21$^{st}$ century, especially in the cold season. Simultaneously, snowfall

decreases (Fig. 4b) markedly in this region,

. Snowfall also  decreases in the Interior Plateau and

Rocky Mountains (Fig. 4d, f), but to a lesser degree than the Coast Mountains

 probably due to persistent cold temperatures at the higher elevations that dominate in this region (Table 1).

Warming temperatures and reduced snowfall induce considerable changes in the snow accumulation and ablation seasons and in snowmelt (Fig. 5). Day-to-day SWE accumulation declines while its seasonality shifts over the 21$^{st}$ century, again with more prominent changes in the Coast Mountains relative to other regions (Fig. 5a). The length of the snow accumulation season is about 38% shorter on average in the 2080s for all geoclimatic regions relative to the 1990s  with a reduction from nearly 80 to 50 days in the Coast (Fig. 5a) and Rocky Mountains (Fig. 5e), and from 65 to 40 days in the Interior Plateau (Fig. 5c). The magnitude and seasonality of snowmelt (Fig. 5b, d, f), which is responsible for generating high flows typically in May or June, shows earlier snowmelt freshets in the future and reduced snowmelt volume. Changes in the partitioning of precipitation between rainfall and snowfall greatly impact the seasonal SWE distribution, consistent with the findings of Islam et al. (2017). While snowmelt during the freshet diminishes overall in the future , unprecedented snowmelt events begin to appear during the cold season in the Coast Mountains (Fig. 5b) by the 2050s, likely due to more frequent warming episodes or perhaps increases in rain-on-snow events.

**3.2 Projected Changes in Runoff Mean, Variability and Seasonality**

4Cthevariabilityaretheprecipitation~~ mean

and  variability of precipitation (Fig. 3a). Consequently, the CMIP5-VIC simulations display larger increases in runoff variability than in mean runoff throughout the simulation period for the entire FRB, Interior Plateau and Rocky Mountains regions. This is not, however, the case in the  Coast Mountains, where the increase in mean runoff (55% by the 2080s) is substantially larger than that in runoff variability (40%). This finding helps to explain why the increase in future runoff is  much more evident in the Coast Mountains region ( Fig. 6).

The substantial increases in cold season runoff are summarized by sub-region and sub-basin in Table 1. Of these, Thompson-Nicola exhibits the largest relative change (+140%) from the 1990s to 2080s, although it is historically the driest of the sub-basins, while the runoff at Hope increases by 71% between the same epochs. With respect to annual runoff, only the Coast Mountains and the Chilko sub-basin display substantial increases (but much smaller in relative terms than cold season increases), with little change elsewhere (Supplementary Table 2). The same qualitative results hold for runoff variability as for means, both in the cold season and annually (Supplementary Table 2). In addition, by the 2080s, the runoff mean and standard deviation more than double over 83% and 71% of the FRB, respectively (Supplementary Fig. 3).

nearly 25 days (consistent with Islam et al., (2017)) in the 2050s and 40 days in the 2080s relative to the 1990s (Fig. 67a). The magnitudes of the annual peak and post-peak flows are, however, progressively diminished in the future periods, with reduced discharge until early October. These changes imply earlier recession to progressively lower flows in summer when salmon are migrating up the Fraser River.

Daily mean runoff in the UF, QU, TN and CH sub-basins exhibits similar features of future change (Supplementary Fig. 32). The advance in the timing of the annual peak flow in these sub-basins is slightly less than for the FRB as a whole (Supplementary Table 2) (~20 days for UF, ~18 days for QU, ~25 days for TN and ~35 days for CH), presumably due to their higher mean elevations. CH features a later freshet (by ~35 days) in the base period compared to the other three sub-basins. This reflects the fact that its flow is partially controlled by the Coast Mountains with influence from the Pacific Ocean along with the presence of large lakes and extensive glaciers in the basin. While the CH sub-basin also features an advance of peak runoff in the future, it exhibits only slightly reduced peak flow magnitudes, unlike the other sub-basins and the FRB as a whole (Supplementary Fig. 32j and Fig. 67a).

In the Rocky Mountains and Interior Plateau along with the UF, QU, TN and LF sub-basins, annual runoff remains stable or increases slightly (Supplementary Table 23) but cold season runoff increases substantially (Table 1). In the Coast Mountains and the CH sub-basin, cold season runoff increases in the future owing to robust rainfall increases year-round. The drainage area mean cold season runoff for the Fraser River at Hope increases from $83\pm2$ mm yr$^{-1}$ in the 1990s to $142\pm10$ mm yr$^{-1}$

~~The changes in daily runoff variability (interannual variability of each day of water year) are modest in summer with small decreases that are consistent with corresponding runoff decreases (Fig. 67b). In contrast, variability increases substantially in the cold season with greater increases in the 2080s than in the 2050s for the Fraser River at Hope (Fig. 67b). Similar changes also emerge in the UF, QU, TN and CH sub basins exhibiting increasing cold season variability with magnitudes comparable to the LF (Supplementary Fig. 32). The changes in daily variability in 7-day moving windows of daily runoff are fairly large in the cold season (Fig. 67c) revealing increased day to day flow fluctuations along with an increase in the interannual variability of daily variability in the 2050s and 2080s.~~

~~The slope is significantly positive between changes in cold season variability and elevations in each gridcell in the Coast Mountains and Interior Plateau geoclimatic regions revealing increasing runoff variability with rising elevations (Supplementary Fig. 3). The Rocky Mountains exhibit strong negative elevation dependency of runoff variability with a 30% reduction in runoff variability with every 1 km increase in elevation during the 2080s.~~

The future evolution of  runoff seasonality shows that while the dominant snowmelt-generated peak flow shifts earlier by ~1 month, noticeable cold season runoff events emerge in winter and spring at the end of the 21$^{st}$ century (Fig. 68). This is most pronounced in the Coast Mountains where fall-winter runoff events rival the summer peak runoff in magnitude (Fig. 8a). The spatially-averaged runoff over the Coast Mountains further highlights the strong increase in cold season peak runoff in this region ( Fig. 4). This increase in cold season peak runoff

magnitude is simulated across the CMIP5-VIC ensemble (Fig. 7c). Apart from the increase in cold season peak runoff magnitude and its annual variability (Supplementary Fig. 75a), the corresponding peak flow occurs somewhat later with warming in the Coast Mountains, moving from late November (~water -day 50) to the beginning of December (~water -day 60) at the end of the 21$^{st}$ century (Supplementary Fig. 4b7b). Overall, as the climate warms, the cold season peak runoff date occurs later in the water year, and thus peak runoff also occurs later. SuchThis increase in cold season peak runoff magnitudes isare simulated by mostacross the CMIP5-VIC simulations ensemble (Supplementary Fig. 4c). Compared to the Coast Mountains, the changes in cold season peak flow timings are much larger in the Interior Plateau and Rocky Mountains probably due to more frequent winter rainfall events on snowpacks (Fig. 4 c and e).

Considering the significant changes in cold season runoff mean and variability, we further employed a multivariate linear regression model to decompose the cold season runoff variability into separate contributions from precipitation and temperature on monthly basis. This procedure allows us to determine the contribution of each key driver to the simulated runoff variability at individual gridcells. Overall, the regression models appear to perform well over whole FRB revealing dominance of temperature driven runoff changes. The multilinear regression model explains 70-90% of the total variance in runoff over a large portion of the basin. The general patterns of runoff trends of the regression model (Figs. 9a) are very similar to that of the MME simulation trends (Figs. 9b). Precipitation contributes to increases in runoff mostly in the Coastal Mountains (Fig. 9c) and over parts of the Rocky Mountains. Contribution of temperature to runoff trends is much higher than precipitation over most of the northern FRB and mountains. For both temperature and precipitation, runoff increase is

[revised manuscript text omitted]

A majority of  gridcells in the Interior Plateau transition from snowmelt-dominant SDC4 to rainfall-dominant SDC1 (33% of gridcells) or SDC2 (26% ) by the end of this century. By contrast,  the Rocky and Coast Mountains show resilience to regime transitions compared to lower elevation regions and remain mostly snowmelt-dominant SDC3  or SDC4  in the 2080s – see Table 32 for details. In the Coast Mountains, the higher elevations resist regime transitioning compared to other elevations that have  robust transitions to rainfall-dominant SDC1 or SDC2 regimes. In contrast with the spatially-averaged response of the geoclimatic regions, routed flows at the outlets of the Upper Fraser, Quesnel, Thompson-Nicola, Chilko and Fraser River at Hope show  a weaker transition of flow regimes (Table 32 and Supplementary Fig. 67). This characteristic arises from the  VIC routing procedure, wherein  model gridcells contributing flows to the outlet  occupy mostly higher elevations and hence

produce flow statistics with more SPs. This attenuation of the climate change signal at channel outlets is consistent with the recent findings of Chezik et al. (2017).

**4 Discussion and Conclusion**

Overall, projected changes in precipitation, SWE and runoff we derived using the CMIP5-VIC modelling chain are in good qualitative agreement with the results of Islam et al. (2017), who used a smaller MME, a different downscaling procedure, and only considered projections up to the 2050s. In this work, we took advantage of a larger MME, an effective downscaling and bias-correction method (BCCAQ2) with proven utility at the daily time scale and a robust snowmelt detection algorithm defining snowmelt-dominant categories (SDCs) to gain a fuller understanding of hydrologic regime change in the FRB throughout the 21$^{st}$ century.

Our results Our results have clarified The overall question driving this study is how future projected increases in air temperature and precipitation mean along with changes in precipitation phase and variability will modulate the FRB's runoff variability and flow regimes through the remainder of this century. suggest that under the projected warming, the changes in precipitation variability and phase, as simulated by CMIP5 models (Pendergrass et al., 2017), play a leading role in declining cold season snowpack accumulation and shifting spring snowmelt earlier in FRB. Thus However, the work raises some additional issues that we address below.To this end, we used VIC simulations, driven by statistically downscaled CMIP5 model projections, to analyse the magnitude, timing and variability of simulated runoff in different sub-basins and geoclimatic regions within the FRB. Furthermore, we used a SP detection algorithm to identify the transition of snowmelt-dominant to rainfall-dominant regimes.

4.1 Projected changes in  and resulting runoff mean

Analysis of contemporary and future climate and hydrological simulations suggests that warming and changes in precipitation variability (Fig. 4) and phase (Fig. 5) will lead to a significant decline in cold season snowpack accumulation and shift spring snowmelt earlier (Fig. 6). The projected change in the runoff mean is sensitive to changes in both the mean and variability of precipitation in a warming climate. These findings are consistent with a recent study by Pendergrass et al. (2017) reporting a global increase of precipitation variability in CMIP5 models simulations on both interannual and daily timescales. projected increases in the precipitation rain-to-snow fraction will have a strong impact on the severity of flooding, for example on mountainsides, with increased spring rainfall accelerating snowmelt runoff.    Despite advances in annual maximum runoff (Fig. 7a), the total annual runoff in most sub-basins does not change substantially (Supplementary Table 2). The stability of total annual runoff implies that a projected increase in rainfall fraction and seasonality will induce considerable changes in runoff timing. The absolute magnitudes of both SWE change and snowmelt are higher in the Coast Mountains (Fig. 6) compared to the Rocky Mountains, likely because the latter region will continue to have cold continental winters with future mean temperatures well below 0°C. Changes in SWE magnitude and seasonality in all three geoclimatic regions shift runoff timing earlier in spring and summer (Fig. 7). Increased cold season precipitation implies more water input to the basin and thus more intense peak flows. In contrast, warmer temperatures reduce the cold season snow and shorten the snow accumulation season hence moderating the snowmelt-driven peak flows in summer.

Annual peak flows in the  FRB's Coast Mountains having a strong maritime influence will shift earlier by around one month by the 2080s (Table 1) and more frequent cold season runoff events will rival

spring freshet flows in magnitude by the end of the 21ˢᵗ century . The source of this enhanced cold season runoff is a topic of ongoing research, but is likely connected to the  increased frequency of landfalling atmospheric rivers simulated in the CMIP5 models  along the North American west coast (Warner et al., 2015; Gao et al., 2015; Payne & Magnusdottir, 2015; Radic et al., 2015; Warner et al., 2015; Warner and Mass, 2017). Under the projected warming, the precipitation phase as rainfall will be a key driver modulating Coast Mountains runoff intensity and frequency especially in the cold season.  Th  may increase the risk of extreme flooding in the Coast Mountains and in the lower Fraser Basin , with associated implications for  water management strategies in these areas.

The  hydrologic response in mountainous regions varies considerably across the basin differentiating its flows mainly into snow-dominant or hybrid (rain and snow) regimes. The SP detection analysis suggest changes in the snow-dominant category arising more prominently across the Interior Plateau probably due to its lower mean elevation with smaller snowpack accumulation in winter

and thus higher sensitivity to temperature increases during the cold season. Snowpack declines are most pronounced at temperatures near freezing that occur more often at Interior Plateau lower elevations during fall or spring. In higher mountainous regions with cold climates, flow regimes depend mainly on the moisture availability and elevation, with higher elevations having cooler air temperatures and thus longer periods of snow accumulation. Therefore the snowpack declines are less sensitive to temperature change in the Rocky Mountains. In the Coast Mountains, the flow regimes will remain rain-dominant at lower elevations owing to abundant rainfall associated with the region's maritime climate.

(flood planning, water storage, etc.), particularly on the southwest coast of BC, will need to be bolstered to cope with projected increases in cold season peak flow events and variability.

Future changes in the FRB's runoff peak flows will also impact salmon life cycles at the stream level. Studies have observed an inverse relationship between salmon spawning survival and higher flows due to the scouring mortality of spawns caused by high runoff in increased rainfall periods (Thorne and Ames 1987; Steen and Quinn 1999; Martins et al., 2012). The projected increase in rainfall and consequently higher flows could increase the scouring mortality of sockeye salmon causing population reductions. Furthermore, the considerable changes in future flows may substantially increase physiological stress during up-river salmon migrations into the FRB and thus further degrade their reproduction rates.

*4.2 Projected changes in runoff variability*

The analysis of the annual cycle of 7-day variability in daily runoff shows evidence of greater day-to-day fluctuations in Fraser River flows in the 2050s and 2080s relative to the 1990s base period (Fig.

67c). The peak day to day variability, occurring mainly in October, intensifies in the future along with a shift towards November. Such increases in the magnitude, variability and shifts in its timing come from projected changes in precipitation phases (Fig. 45) and increases in cold season snowmelt events (Fig. 56). The projected changes in day to day flow variability in the summer probably arise from changes in the magnitude and timing of the spring freshet. Accelerated and earlier snowmelt may result in flashier flows causing the increase for potential flooding or attenuated spring freshet with attendant drought potential in summer. Furthermore, grid scale projected changes in runoff variability are strongly dependent on the elevation (Supplementary Fig. 73). Higher elevations in the Coast Mountains will experience greater changes in runoff variability compared to lower elevations of the Interior Plateau. The main factors controlling cold season runoff variability are the changes in temperature mean and precipitation means and variability. This suggests that precipitation variability will contribute to more intense cold season runoff variability (Fig. 4) in addition to any increase that may be due to higher temperatures.

**4.43 Transitions between runoff regimes**

The FRB exhibits substantial spatial variation of air temperature and precipitation due to its complex topography and maritime influences. The hydrologic response therefore varies considerably across the basin differentiating its flows mainly into snow-dominant or hybrid (rain and snow) regimes. Based on the results of the SP detection analysis, the FRB will transition to a less snowmelt-dominated regime in the 21[st] century. The Interior Plateau will transition from snow-dominant to increasingly rain-dominant regimes (Table 2 and Fig. 10). Changes in the snow-dominant category arise more prominently across the Interior Plateau owing to its lower mean elevation with smaller snowpack

accumulation in winter and thus higher sensitivity to changes in air temperature during the cold season. Snowpack declines are most pronounced at temperatures near freezing that occur more often at Interior Plateau lower elevations during fall or spring.

In FRB mountainous regions, flow regimes depend mainly on the proximity to marine influences and elevation, with higher elevations having cooler air temperatures and thus longer periods of snow accumulation. Therefore the snowpack declines are less sensitive to temperature change in the Rocky Mountains with a possibility of experiencing increased snowfall due to higher moisture availability.

Future percentages of SP years for all sub-basins are considerably larger than the spatially-averaged values of the geoclimatic regions overlapping them (Table 2). The former, computed at basin outlets, are sensitive to the channel network and the routing model used to route these sub-basin flows. In the routing procedure, the model gridcells contributing flows to the outlet gauge occupy mostly higher elevations and hence produce flow statistics with more SPs. This attenuation of the climate change signal at channel outlets is consistent with the recent finding of Chezik et al. (2017) reporting river networks dampen local hydrologic signals of climate change since the regional characteristics of the channel network aggregate the upstream climate at different spatial scales.

***Sensitivity of Results to Other Forcings:*** This study deals solely with the impacts of projected changes in climate on the FRB's cold season runoff variability and flow regimes under strong greenhouse gas (GHG) forcing and does not investigate the projected land cover change on regional hydrology. Influences of other forcings, such as land cover change and glacier growth or loss, are neglected, similar to other recent modelling studies of projected climate change impacts in

the FRB (Schnorbus et al., 2014, Shrestha et al., 2012 and Islam et al., 2017). Although climate change may influence future forest dynamics (e.g., Marlon et al., 2009; Gonzalez et al., 2010),   We point out that any choice of land use/land cover scenario is arbitrary considering that all future changes are conditional on a given GHG scenario. In addition, it is arguably more difficult to projct how forest composition and disturbance regimes, which are regional by nature, will change in the future compared to globally well-mixed GHG concentrations (although we acknowledge that we have not explored the full range of available RCPs either). Several previous studies have shown that the sensitivity of runoff to forest cover change depends on a basin's size and regional characteristics (Wei et al., 2013; Zhang et al., 2017). The forest cover change response generally decreases with increasing basin size, with large snow-dominated basins being more resilient. Limited support for this is found in the study of Schnorbus et al. (2010), who utilized VIC simulations to quantify the impacts of idealized scenarios of mountain pine beetle and associated salvage harvesting across different watersheds within the FRB. They found that despite a large upstream sensitivity to land cover changes, the overall, integrated change in discharge at Hope, BC was quite low. ~~Focusing In the Willow River basin within the Interior Plateau of the FRB, Wei and Zhang (2010) found a ~20% increase in runoff even after an increase in clear-cut area fraction of up to 80%.Hence, we suspect that thydrology in modest,sLimited support for this is found in the~~

 ~~The sensitivity of runoff to forest cover change depends on the basin's size and its regional characteristics (Wei et al., (2013); Zhang et al., 2017). The forest cover change response decreases with increasing size of the large basins with large snow dominated basin being more resilient. However the forest cover is more sensitive in water-limited watersheds than in energy-limited watersheds (Zhang et al., 2017). Focusing the Willow River basin within Interior Plateau, Wei and Zang, (2010) found ~20% increase in the runoff even after the increase in clear cut area fraction up to 80%. Havel et al., (2018) quantified the wildfire influence on streamflow in mountainous catchments and found even large runoff changes generated by high burnt area fractions in small sub-catchments have a relatively small influence on cumulative runoff in the larger parent watershed.~~

[revised manuscript text omitted]

Martins, E. G., Hinch, S. G., Cooke, S. J. and Patterson, D. A.: Climate effects on growth, phenology and survival of sockeye salmon (Oncorhynchus nerka): a synthesis of the current state of

knowledge and future research directions. Reviews in Fish Biology and Fisheries, 22, 887-914, 2012.

Maurer, E. P., Hidalgo, H. G., Das, T., Dettinger, M. D., and Cayan, D. R.: The utility of daily large-scale climate data in the assessment of climate change impacts on daily streamflow in California, Hydrol. Earth Syst. Sci., 14, 1125-1138, doi:10.5194/hess-14-1125-2010, 2010.

McAdam, S. O., Walters, C. J., and Nistor, C.: Linkages between white sturgeon recruitment and altered bed substrates in the Nechako River, Canada, Trans. Amer. Fish. Soc., 134, 1448-1456, 2005.

Moore, R. D.: Hydrology and water supply in the Fraser River Basin. In water in sustainable development: exploring our common future in the Fraser River Basin, AHJ Dorcey, JR Griggs (eds). Wastewater Research Centre, The University of British Columbia: Vancouver, British Columbia, Canada; 21–40, 1991.

Morrison, J., Quick, M. C. and Foreman, M. G. G.: Climate change in the Fraser River watershed: flow and temperature projections, J. Hydrol., 263(1–4), 230–244, doi:10.1016/S0022-1694(02)00065-3, 2002.

Nash, J. E. and Sutcliffe, J. V.: River flow forecasting through conceptual models part I - A discussion of principles, J. Hydrol., 10(3), 282–290, doi:10.1016/0022-1694(70)90255-6, 1970.

fftcmNijssen, B., O'Donnell, G. M., Lettenmaier, D. P., Lohmann, D. and Wood, E. F.: Predicting the discharge of global rivers, J. Climate, 14(15), 3307–3323, doi:10.1175/1520-0442(2001)014<3307:PTDOGR>2.0.CO;2, 2001b.

[revised manuscript text omitted]

---

## Author Response (AR2)

**Responses to the Editor and Reviewer**

*We provide our responses (in bold italic) to the comments submitted by the Editor and Reviewer #1 below.*

**Editor:**

Thank you for addressing the reviewer comments and suggestions. The paper provides some interesting results and has been much improved. However, one of the reviewers still has major concerns about the significance and novel contribution of this paper (from an international perspective). There is still some room for improvement, as noted and explained by reviewer #1. That would help make this more than another mere case study, which had been a concern of the reviewers from the start. Please address this and engage more meaningfully with the international literature, especially in the introduction and the discussion.

***Thank you for your decision on our manuscript and for providing this additional feedback on our work. In regards to the comments from Reviewer #1, we have further revised the introduction and study domain sections in our manuscript by including a literature review that includes a broader, international scope. We have explicitly mentioned how this paper covers the knowledge gaps in the available literature on projected flow variability and hydrological regime transitions. We also made some minor text adjustments in the abstract and discussion sections.***

**Reviewer # 1**

As I noted in my review of the original manuscript, I have not been following the literature on climate change effects on hydrology for a few years, and am not qualified to address many of the detailed aspects of the analysis. Instead, this review will focus on a high-level assessment of the suitability of the revised manuscript for publication in a leading international journal. The authors have made a number of revisions to the manuscript in response to the review comments, including the addition of some new analysis. However, the focus of the presentation remains the Fraser River basin (FRB), and it is still unclear to me what is new and significant from the perspective of an international reader. The authors argue that the FRB has the unique attribute that it spans a range of hydroclimatic contexts. This point may be valid, but it does not establish the international value of studying its response to climatic warming. It seems to me that the same lessons could be generated by synthesizing the results of individual studies on a range of currently nival catchments. For example, I would expect the North Cascades in Washington State to respond similarly to the Coast Mountains in British Columbia. Like the FRB, the Rhine River basin includes a broad range of hydroclimatic regimes.

In my opinion, if the authors want to publish this work in an international journal, they will need to overhaul the introduction, most of which is currently a description of the geoclimatic characteristics of the FRB and a review of previous studies focused on it. The authors will need to provide a careful review of the international literature to identify what we currently know about how catchments with nival streamflow regimes will transition toward a greater pluvial influence under climatic warming scenarios. This delineation of current knowledge would then provide a framework for identifying specific, internationally significant gaps in knowledge that,

in turn, could be used to generate specific research questions or hypotheses to guide the analysis and interpretation.

*Thank you for reviewing this manuscript. As per your suggestions, the introduction section has been updated considerably with examples from the international literature focusing on climate change impacts on several nival basins. In this regard, we have included three new paragraphs in the introduction section that highlight the knowledge gaps in the existing literature and what new and significant information is provided in our paper.*

*Our paper, however, still emphasizes the Fraser River Basin given the study is designed for the HESS special issue on "Understanding and predicting earth system and hydrological change in cold regions" that has a regional focus on western Canada (the description of the HESS special issue is copied below). The Fraser River Basin spans 240,000 km², an area equivalent to that of the United Kingdom, and exhibits hydrological characteristics that typify cold regions in its diverse landscapes from the relatively dry Interior Plateau, to the snowy Rocky Mountains and the maritime-influenced Coast Mountains. We strongly believe that our manuscript is well aligned with the scope of the HESS special issue by providing a counterpart to studies focused on the Mackenzie and Saskatchewan River Basins on the eastern side of the western Cordillera's continental divide. It also provides a sister study to the paper on peak annual flows in the Fraser River Basin by Curry and Zwiers (2018) that was also published in this special issue of HESS.*

*"Description of the Special Issue:*
*Statement of purpose of the special issue: Global warming and other climatic changes are causing rapid and widespread changes to the landscape, vegetation, and water cycling. Cold regions, such as the interior of western Canada, are at the forefront of this change and are highly sensitive to further warming. Dramatic changes are occurring to landscapes and hydro-ecological systems. There is therefore an urgent need to understand the nature of the changes and to develop the improved modelling tools needed to manage uncertain futures. To this end, the Changing Cold Regions Network (CCRN; www.ccrnetwork.ca) aims to understand, diagnose, and predict interactions amongst the cryospheric, ecological, hydrological, and climatic components of the changing Earth system at multiple scales with a geographic focus on western Canada, including the Saskatchewan and Mackenzie River basins. CCRN is a Canadian-led research programme led by a consortium of eight universities and four federal agencies, due to be completed in March 2018. It is therefore timely for a special issue to bring together papers that synthesize this research and address recent advances in understanding, diagnosis, and prediction of past and future changes in cold-region Earth systems either as part of the CCRN initiative or from other studies around the world. The development and use of numerical models to diagnose past change and predict future sensitivity and response under various climate and land-cover scenarios is a particular focus. Key questions of relevance include whether cold-region hydrological processes and their interactions have changed in response to climatic drivers, what the feedbacks and thresholds leading to cold-region Earth system changes are, and/or what factors impart hydrological resilience or sensitivity to change in cold regions. The special issue is open to all submissions within its scope and welcomes related studies from cold-region environments around the world".*

Based on the authors' responses to reviewers, it appears that the role of atmospheric rivers is one of the phenomena of interest in this study. If this is the case, then the authors should make this phenomenon a focus of the literature review in the introduction. In both the revised manuscript and the responses to the review comments, the authors address atmospheric rivers as a primarily west coast North American phenomenon, whereas atmospheric rivers and their hydrologic effects have been addressed in a global context. See, for example, Espinoza et al. (2018).

***We have now included a paragraph in introduction section focusing the literature review on atmospheric rivers and their importance for the west coast of North America and the Fraser Basin, in particular.***

**Reference:**
Charles L. Curry and Francis W. Zwiers: Examining controls on peak annual streamflow and floods in the Fraser River Basin of British Columbia, Hydrol. Earth Syst. Sci., 22, 2285-2309, https://doi.org/10.5194/hess-22-2285-2018, 2018.

[revised manuscript text omitted]

---

## Author Response (AR3)

**Responses to the Editor**

**We provide our responses (in bold italic) to the additional comments submitted by the handling Editor, Dr. Chris DeBeer, below.**

**Editor:**

This paper has been improved and is nearly ready for final publication. However, I still have a couple of minor concerns. 1) With regard to how land cover was handled in the modelling, it is a limitation of the study that this was treated as static. The point is made that any choice of land cover scenario is arbitrary considering changes depend on a GHG scenario. The study uses the RCP8.5 emissions scenario, and under this is scenario it is understood (with much literature to support this) that there will be a dramatic increase in the occurrence and magnitude of wildfires. Not to mention other disturbances. The suggestion that this would not have a major impact on flow at the outlet of the basins is weakly supported and quite speculative. 2) With regard to the treatment of glaciers, the model handles these in a crude manner and these are again treated as static. The introduction section discusses how the loss of glaciers and other storage is expected to lead to increased inter-annual variability of flow in such basins, then the discussion dismisses this, notably without supporting evidence from this study. Perhaps the climatic forcing is a much more important driver of changes in flow regime than glacier loss, but this study did not examine this. 3) The modelling approach in this study did not allow for an in-depth consideration of the dynamic aspects of hydrological functioning and response to be represented (i.e. changing flow pathways, process interactions, feedbacks, etc.). These might have important implications across different scales.

I point these out as some limitations of the study. I would suggest that these be more clearly acknowledged in the discussion and conclusion, perhaps setting the context for further investigation.

Thank you for your decision on our revised manuscript (MS No: hess-2018-232) and for your additional comments regarding the VIC model limitations. To incorporate your suggestions, we have further revised the discussion section in our manuscript by editing the text about the impact of land use and glacier changes on projected streamflow in the Fraser River Basin (Page 23, Lines 3-21 and Page 24, Lines 1-5). We have also incorporated some minor text edits throughout the manuscript and in the conclusions, have reminded the reader of the discussion of the model limitations. In addition, Figure 3 in the manuscript is replotted with some formatting to the sub panels.

We agree with your concern that the VIC model used in the present study cannot fully address both land use and glacier changes in its simulations. In our future efforts, we have plans to undertake sensitivity tests for these effects in hydrological simulations. Our research team is currently focusing on the development and application of an updated and re-engineered version of the VIC model (VIC-GL) that is coupled with snow mass and glacier dynamics models. Our future studies will therefore utilize the VIC-GL model to quantify the dynamic contribution of glacier melt to runoff in several basins in western Canada including the FRB. We have now explicitly mentioned this in the discussion section of the manuscript (Page 24, Lines 1-5).

Regarding your comment about in-depth consideration of the dynamic aspects of hydrological functioning and feedbacks, we point out that the VIC model already accounts for many important process interactions, while feedbacks from VIC back onto the climate are not permitted by the experimental design (which is common with other similar studies). With regard to changing soil water flow pathways under future warming, we suspect that this might be a larger concern in the context of permafrost degradation in an Arctic setting or in high mountainous terrain. In a region such as the Fraser River Basin, however, changes in the soil drainage pathways are presumably much less of a concern.

We hope that this response and the additional modifications made to the manuscript sufficiently address your concerns.

[revised manuscript text omitted]
 -1 ]           1990s (1980-2009)         2050s (2040-2069)         2080s (2070-2099) |             |             |             |             |             | Advance
(days) by |
|---------------------------------------|--------------------|-------------------|-------------------|---------------------------------------------------------------------------------------------------------------------------------------------------------------|-------------|-------------|-------------|-------------|-------------|----------------------|
| [WSC ID]                              | [km 2 ] | Longitude
[°W] | [ m ]      | Mean                                                                                                                                                          | Variability | Mean        | Variability | Mean        | Variability | 2080s                |
| Rocky
Mountains                    | -                  | -                 | 1567              | 160 ± 3                                                                                                                                                       | 25 ± 2      | 205 ± 12    | 51 ± 7      | 270 ± 22    | 65 ± 9      | -                    |
| Interior
Plateau                   | -                  | -                 | 1101              | 58 ±
0                                                                                                                                                     | 10 ± 1      | 78 ±
4   | 19 ± 2      | 102 ± 7     | 22 ± 2      | -                    |
| Coast
Mountains                    | -                  | -                 | 1296              | 450 ± 10                                                                                                                                                      | 90 ± 7      | 606 ± 23    | 139 ± 11    | 730 ± 34    | 158 ± 15    | -                    |
| Upper Fraser
(UF)
[08KB001]     | 32400              | 54.01,
122.62  | 1308              | 76 ±
8                                                                                                                                                     | 29 ± 4      | 107 ±
13 | 46 ± 7      | 133 ±
18 | 54 ± 9      | 20                   |
| Quesnel (QU)
[08KH006]             | 11500              | 52.84,
122.22  | 1173              | 119±
3                                                                                                                                                     | 24 ± 2      | 142 ± 8     | 42 ± 6      | 182 ±
16 | 52 ± 7      | 18                   |
| Thompson-
Nicola (TN)
[08LF051] | 54900              | 50.35,
121.39  | 1747              | 47 ±
2                                                                                                                                                     | 13 ± 1      | 74 ±
7   | 28 ± 5      | 113 ±
12 | 38 ± 4      | 25                   |
| Chilko (CH)
[08MA001]              | 6940               | 52.07,
123.54  | 1756              | 95 ±
2                                                                                                                                                     | 20 ± 2      | 119 ±
6  | 34 ± 5      | 151 ±
10 | 42 ± 4      | 35                   |
| Fraser at Hope
(LF)
[08MF005]   | 217000             | 49.38,
121.45  | 1330              | 83 ±
2                                                                                                                                                     | 13 ±1       | 109 ±
6  | 26 ± 4      | 142 ± 10    | 33 ±4       | 25                   |

**Table 2:** Decomposition of the key drivers affecting cold season runoff changes in the 2080s. Contributions are in % estimated using the multivariate linear regression (MLR, described in section 2.4.2) model using monthly time series from 21 CMIP5 simulations.  $R^2$  provides the variance explained by all three variables. The gridcell averaged contributions are estimated only for statistically significant values at p < 0.05. Change in each variable is normalized by the total runoff change to estimate its contribution. The contributions listed in last three columns do not necessarily sum exactly 100% due to rounding off and/or masking of insignificant gridcells before taking area-averages.

| Region             | Runoff
2080 | Change
s (%) | R 2 | Contribution to Runoff Change (%) |          |                                |  |
|--------------------|----------------|-----------------|----------------|-----------------------------------|----------|--------------------------------|--|
|                    | VIC            | MLR             |                | Rainfall                          | Snowfall | Mean Air
Temperature |  |
| Coast Mountains    | 121            | 146             | 0.70           | 61                                | 8        | 32                             |  |
| Interior Plateau   | 114            | 140             | 0.42           | 44                                | 4        | 46                             |  |
| Rocky Mountains    | 79             | 117             | 0.45           | 44                                | 6        | 50                             |  |
| Fraser River Basin | 107            | 133             | 0.45           | 47                                | 8        | 46                             |  |

**Table 3:** CMIP5-VIC simulated projected change in snowmelt-dominant regimes. Values are in % calculated using the ratio of the sum of SPs years and the total number of years over all 21 CMIP5-VIC simulations. The uncertainty ranges  $(\pm)$  indicate inter-model spread in the MME mean values indicated by a 5-95% models range.

| Region           | Snowmelt-Dominant Regime (%) |                   |                   |  |  |  |
|------------------|------------------------------|-------------------|-------------------|--|--|--|
| Region           | 1990s (1980-2009)            | 2050s (2040-2069) | 2080s (2070-2099) |  |  |  |
| Coast Mountains  | 95±1                         | 74±3              | 57±4              |  |  |  |
| Interior Plateau | 100±0                        | 73±5              | 43±7              |  |  |  |
| Rocky Mountains  | 100±0                        | 88±3              | 64±6              |  |  |  |
| UF               | 95±9                         | 95±9              | 93±9              |  |  |  |
| QU               | 100±0                        | 98±1              | 85±5              |  |  |  |
| TN               | 100±0                        | 98±1              | 80±8              |  |  |  |
| СН               | 100±0                        | 100±0             | 95±1              |  |  |  |
| LF               | 100±0                        | 98±1              | 81±7              |  |  |  |